# Statistical Undecidability in Linear, Non-Gaussian Causal Models in the Presence of Latent Confounders

**Konstantin Genin**[*]
University of Tübingen
konstantin.genin@gmail.com

## Abstract

If causal relationships are linear and acyclic and noise terms are independent and Gaussian, causal orientation is not identified from observational data — even if faithfulness is satisfied (Spirtes et al., 2002). Shimizu et al. (2006) showed that acyclic, linear, **non**-Gaussian (LiNGAM) causal models *are* identified from observational data, so long as no latent confounders are present. That holds even when faithfulness fails. Genin and Mayo-Wilson (2020) refine that result: not only are causal relationships identified, but causal orientation is *statistically decidable*. That means that for every $\epsilon > 0$, there is a method that converges in probability to the correct orientation and, at every sample size, outputs an incorrect orientation with probability less than $\epsilon$. These results naturally raise questions about what happens in the presence of latent confounders. Hoyer et al. (2008) and Salehkaleybar et al. (2020) show that, although the causal model is not uniquely identified, causal orientation among observed variables is identified in the presence of latent confounders, so long as faithfulness is satisfied. This paper refines these results: although it is possible to converge to the right orientation in the limit, causal orientation is no longer statistically decidable—it is not possible to converge to the correct orientation with finite-sample bounds on the probability of orientation errors, even if faithfulness is satisfied. However, that limiting result suggests several adjustments to the LiNGAM model that may recover decidability.

## 1 Introduction

Spirtes et al. [2000] develop the elements of causal discovery from observational data in the linear Gaussian setting. They show that when functional relationships between variables are linear and acyclic and noise terms are independent and Gaussian, it is possible to converge to the Markov *equivalence class* of the graph generating the data. Although some non-trivial causal information can be recovered, causal orientation is in general not identified: two linear Gaussian models may differ in causal orientation and nevertheless generate the exact same distribution over the observed variables. Identifiability fails even when causal faithfulness is satisfied and no hidden variables are present. For this reason, it was a significant advance when Shimizu et al. [2006] showed that, when functional relationships between variables are linear and acylic and noise terms are independent and *non*-Gaussian, all causal orientations can be uniquely identified from observational data, even without assuming faithfulness. However, that early result depends on the absence of unobserved confounders. Since then, the LiNGAM framework, as it came to be called, has been extended to accommodate the presence of hidden variables [Hoyer et al., 2008, Salehkaleybar et al., 2020]. Salehkaleybar et al. [2020] prove that if, in addition to the usual LiNGAM assumptions, we assume causal faithfulness, then causal ancestry relationships between observed variables are identified even in the presence of

---

[*]Research Group Leader "Ethics and Epistemology of Machine Learning" at the Cluster of Excellence "Machine Learning for Science", University of Tübingen.

35th Conference on Neural Information Processing Systems (NeurIPS 2021).

Table 1: Three Varieties of Decidability

| | | then the output is probably . . . | | |
| --- | --- | --- | --- | --- |
| | | correct[†] at all $N$ | informative after known $N$ | correct and informative after unknown $N$ |
| When orientation is . . . | Uniformly Decidable | ✓ | ✓ | ✓ |
| | Decidable | ✓ | X | ✓ |
| | Decidable in the limit | X | X | ✓ |

† Outputs expressing suspension of judgement are here considered correct.

unobserved confounders. In other words: if two faithful, confounded LiNGAM models generate the same distribution over the observed variables, then for every pair of observed variables $X, Y$, the models must agree on whether $X$ is causally upstream of $Y$, $Y$ is upstream of $X$, or neither is upstream of the other. Note that the models do not have to agree on which variables are *direct* causes of which others, only on which variables are *ancestors* of which others. Moreover, although all models generating the same distribution over the observed variables must agree on the causal ancestry relations between them, they may disagree on the strength of the causal effects (Figure 2).

These identifiability results are exciting theoretical developments. However, identifiability is a weak criterion and on its own does not entail the existence of a consistent discovery algorithm. Moreover, distinctions ought to be made between degrees of identifiability. For example, *uniform* decidability (elsewhere known as uniform consistency) requires that one be able to determine *a priori* a sample size at which the chance of identifying the true orientation is at least $1 - \alpha$, no matter which causal model is generating the data. Unfortunately, it is easy to show that there is no uniformly consistent algorithm for determining the direction of a causal edge, even in unconfounded LiNGAMs (see Example 1 in Genin and Mayo-Wilson [2020]). Although one could strengthen the LiNGAM assumptions to make uniform decidability feasible, these assumptions would probably be implausibly strong .[2]

Decidability *in the limit* (elsewhere known as pointwise consistency) requires only that for each causal model that might be generating the data, there be some sample size by which the chance of outputting the true orientation is at least $1 - \alpha$. Note that this is compatible with all kinds of short-run behavior. For example, Kelly and Mayo-Wilson [2010] show that in the unconfounded linear Gaussian setting, even in situations in which causal orientation is identifiable, it is possible to force any consistent discovery procedure to "flip" its judgement about whether $X$ causes $Y$ or vice versa. That means that for any such procedure, it is possible to find a model in which the procedure flips between being highly likely to output the correct causal orientation at a smaller sample size and highly likely to output the incorrect orientation at a larger sample size, with the number of such flips bounded only by the number of variables in the model.

Genin and Mayo-Wilson [2020] introduce a new notion of decidability that is intermediate between uniform decidability and decidability in the limit. Statistical decidability requires that a discovery procedure converge to the true orientation as sample sizes increase and, at every sample size, output an incorrect orientation with probability at most $\alpha$. Note that statistical decidability rules out flipping behavior, but is consistent with suspending judgement at arbitrarily large sample sizes. Genin and Mayo-Wilson [2020] show that, in the absence of unobserved confounders, causal orientation is decidable in the LiNGAM setting and therefore, flipping can be avoided. That demonstrates that statistical decision procedures may exist even when uniform decision procedures do not.

---

[2]In the linear Gaussian setting, Robins et al. [2003] show that uniform consistency is not feasible under standard assumptions. Zhang and Spirtes [2003] propose a strengthening of these assumptions to ensure uniform decidability, but Uhler et al. [2013] argues that this rules out a topologically large set of models. Bühlmann et al. [2014] give conditions strong enough the ensure uniform decidability in the general setting of additive structural equation models.

Table 1 sums up: if we have a *uniform* decision procedure we know that the output of the method is probably correct at every sample size and we know *a priori* how large a sample we need for it to be informative. If we have a decision procedure, we know that the output of the method is probably correct at every sample size, but we do not know *a priori* how large a sample we need for it to be informative. And if we have only a limiting decision procedure, we can never be sure that the output is correct, although we know it will be probably correct eventually.

Crucially, mere identifiability alone does not entail that a problem is decidable in any of the above three senses. Therefore, the results of Salehkaleybar et al. [2020] leave open the possibility that the causal ancestry relationship is not even decidable in the limit. The first main result of this paper is that the ancestry relationship is indeed decidable in the limit—there exist consistent procedures for learning the causal ancestry relationship between observed variables. Unfortunately, flipping returns in the confounded LiNGAM setting, even when we assume causal faithfulness. Although consistent procedures exist for learning causal orientation, consistent *decision* procedures do not. Table 2 sums up these results. However, that result suggests several adjustments to the LiNGAM framework that could recover decidability. These are discussed in Section 7.

Table 2: Causal Orientation in LiNGAM models

|  | **Unconfounded** | **Potentially Confounded** |
|---|---|---|
| **Faithful** | decidable | decidable in the limit |
| **Unfaithful** | decidable | not identified |

Genin and Mayo-Wilson [2020] prove the results in the first column.
The result in the upper-right cell is proven in this paper.

## 2   Technical Preliminaries

We first introduce notation for manipulating matrices. Suppose $A$ is an $n \times p$ matrix and $U, V$ are subsets of $\{1, \ldots, n\}, \{1, \ldots, p\}$, respectively. Let $A_{[U;V]}$ be the result of dropping all rows from $A$ that are not in $U$ and all columns that are not in $V$. Let $A_{(U;V)}$ be the result of dropping all rows from $A$ that are in $U$ and all columns that are in $V$. We write $A_{(i,j)}$ for $A_{(\{i\};\{j\})}$ and $A_{ij}$ for $A_{[\{i\};\{j\}]}$. We say $A$ has **pairwise linearly independent** columns iff no two columns of $A$ are proportional.

Let $\mathcal{M}$ be a set of statistical models. We assume there is a function $P : M \mapsto P_M$ that maps each model in $\mathcal{M}$ to a probability measure over a space $\Omega$ of observed outcomes, although we often do not distinguish between a random vector and the probability measure induced by its distribution function. Henceforth, we assume $\Omega = \mathbb{R}^p$. We lift $P(\cdot)$ to sets of models in the obvious way: if $\mathcal{A} \subseteq \mathcal{M}$, let $P[\mathcal{A}] = \{P(M) : M \in \mathcal{A}\}$. Let $\mathcal{P} = P[\mathcal{M}]$. Let $\mathcal{P}_0 \subseteq \mathcal{P}$ be the probability measures in $\mathcal{P}$ absolutely continuous with Lebesgue measure on $\Omega$. Let $P_0[\mathcal{M}] = \mathcal{P}_0 \cap P[\mathcal{M}]$. If $A \subseteq \mathbb{R}^{nd}$, let $\partial A$ be the boundary of $A$ in the usual topology on $\mathbb{R}^{nd}$. The **weak topology** on $\mathcal{P}$ is defined by letting a sequence of Borel measures $P_n$ converge weakly to $P$, written $P_n \Rightarrow P$ iff $P_n(A) \to P(A)$, for every $A$ such that $P(\partial A) = 0$. A collection of random vectors $(\mathbf{X}_n)$ converges in distribution to $\mathbf{X}$ iff the probability measures induced by the $\mathbf{X}_n$ converge weakly to the measure induced by $\mathbf{X}$. We write $\mathrm{cl}(\cdot)$ for the closure operator in the weak topology. We say that a set is **locally closed** iff it is the intersection of an open and a closed set. Although every open set and every closed set is also locally closed, the converse is not true: there are properly locally closed sets which are neither open nor closed. In metrizable topologies such as the topology of weak convergence, every open set, and therefore every locally closed set, is a countable union of closed sets. For any natural number $k$, let $P_M^k$ be the $k$-fold product measure of $P_M$ with itself. This measure describes the probabilities of events in $\mathbb{R}^{kd}$ when we take $k$ iid samples from $P_M$. If the measures $P_n$ converge weakly to $P$, the product measures $P_n^k$ also converge weakly to $P^k$ (see Theorem 2.8 in Billingsley [1986]).

We define a **question** $\mathfrak{Q}$ to be a countable set of disjoint subsets of $\mathcal{M}$. The elements of $\mathfrak{Q}$ are called **answers**. For all $M \in \cup \mathfrak{Q}$, let $\mathfrak{Q}(M)$ denote the unique answer in $\mathfrak{Q}$ containing $M$. The answer to question $\mathfrak{Q}$ is **identified** iff $P(M) \neq P(M')$ whenever $\mathfrak{Q}(M) \neq \mathfrak{Q}(M')$. Given a question $\mathfrak{Q}$, we define a **method** $\lambda = \langle \lambda_n \rangle_{n \in \mathbb{N}}$ to be a sequence of measurable functions $\lambda_n : \Omega^n \to \mathfrak{Q} \cup \{\mathcal{M}\}$, where $\lambda_n$ maps samples of size $n$ to answers to the question; a method may also take the value $\mathcal{M}$ to indicate that the data do no fit any particular answer sufficiently well, and so we call $\mathcal{M}$ the **uninformative answer**. We require that $\partial \lambda_n^{-1}(\mathcal{A})$ has Lebesgue measure zero for all $n$ and every answer $\mathcal{A}$ in the range of $\lambda_n$.

A method is (pointwise) **consistent** for $\mathcal{Q}$ if for all $\epsilon > 0$ and $M \in \cup\mathfrak{Q}$, there is $n$ such that $P_M^k(\lambda_k = \mathfrak{Q}(M)) > 1 - \epsilon$ for all $k \geq n$. We say that $\mathfrak{Q}$ is **decidable in the limit** iff there is a consistent method for $\mathfrak{Q}$. Dembo and Peres [1994] give the following sufficient condition for limiting decidability. For a proof, see their Corollary 2.

**Theorem 2.1** (Dembo and Peres [1994]). *$\mathfrak{Q}$ is decidable in the limit if $\{P(A) : A \in \mathfrak{Q}\}$ is disjoint and each $P(A)$ is a countable union of sets closed in the weak topology.*

Given some $\alpha > 0$, say that a method $\lambda$ is an $\alpha$-**decision procedure** for $\mathfrak{Q}$ if (1) $\lambda$ is consistent for $\mathfrak{Q}$ and (2) $P_M^n(M \notin \lambda_n) \leq \alpha$ for all $M \in \cup\mathfrak{Q}$ and all sample sizes $n$. In other words: an $\alpha$-decision procedure outputs a false hypothesis with probability at most $\alpha$. A question is **statistically decidable** (or simply decidable) if there is an $\alpha$-decision procedure for $\alpha > 0$. We give a simple necessary condition for statistical decidability.

**Theorem 2.2.** *$\mathfrak{Q}$ is statistically decidable only if $P_0(\mathcal{A}) \cap \mathsf{cl}P(\mathcal{B}) = \varnothing$ for all $\mathcal{A}, \mathcal{B} \in \mathfrak{Q}$.*

*Proof of Theorem 2.2.* Suppose for a contradiction that $P_M \in P(\mathcal{A}) \cap \mathsf{cl}(P(\mathcal{B}))$ is absolutely continuous with Lebesgue measure and $\lambda$ is an $\alpha$-decision procedure for $\mathfrak{Q}$. Since $\lambda$ is consistent for $\mathfrak{Q}$, there must be $n$ such that $P_M^n(\lambda_n^{-1}(\mathcal{A})) > 1 - \alpha$. Since $\partial\lambda_n^{-1}(\mathcal{A})$ has Lebesgue measure zero and $P_M^n$ is absolutely continuous with Lebesgue measure on $\mathbb{R}^{np}$, $P_M^n(\partial\lambda_n^{-1}(\mathcal{A})) = 0$.[3] Therefore, there are $(M_i)$ in $\mathcal{B}$ such that $P_{M_i}^n(\mathcal{A}) \to P_M^n(\mathcal{A})$. But then there is some $M_j \in \mathcal{B}$ such that $P_{M_j}^n(M_j \notin \lambda_n) > \alpha$. Contradiction. $\square$

It is worth introducing some intuitive language for questions $\mathfrak{Q}$ with only one (usually non-exhaustive) answer $\mathcal{A} \subseteq \mathcal{M}$. We say that $\mathcal{A}$ is **statistically verifiable** iff $\mathfrak{Q} = \{\mathcal{A}\}$ is decidable. Say that $\mathcal{A}$ is **statistically refutable** iff $\mathfrak{Q} = \{\mathcal{M} \setminus \mathcal{A}\}$ is decidable. For partial converses of Theorems 2.1 and 2.2, see Genin and Kelly [2017]. Essentially, the converses hold straightforwardly if all distributions are assumed absolutely continuous with Lebesgue measure.

The fundamental result of this paper is that the question of whether (observed) $X$ is an ancestor of (observed) $Y$ or vice-versa, while decidable in the limit, is not decidable. The former follows from Corollary 6.7 and an appeal to Theorem 2.1. The latter is proven by showing that you can approximate to an arbitrary degree of precision the distribution over $(X, Y)$ generated by a faithful LiNGAM in which $X \to Y$ by a sequence of distributions $(X_m, Y_m)$ generated by faithful but confounded LiNGAMs in which $X_m \leftarrow Y_m$ (see Figure 3 and Lemma 6.2). Undecidability follows by appeal to Theorem 2.2.

## 3   Acyclic Linear Causal Models

An **acyclic linear causal model in** $d$ **variables**[4] $M$ is a triple $\langle \mathbf{X}, \mathbf{e}, A \rangle$, where $\mathbf{X} = \langle X_i \rangle$ is a vector of $d$ random variables, $\mathbf{e} = \langle e_1, e_2, \ldots, e_d \rangle$ is a random vector of $d$ exogenous noise terms, and $B$ is a $d \times d$ matrix such that

1. Each variable $X_i$ is a linear function of variables earlier in the order, plus an unobserved noise term $e_i$:
$$X_i = \sum_{j<i} A_{ij} X_j + e_i;$$

2. the noise terms $e_1, \ldots, e_d$ are mutually independent.

In matrix notation, we have that $\mathbf{X} = A\mathbf{X} + \mathbf{e}$. Because no $X_i$ causes itself, $A$ has only zeroes along its diagonal. By virtue of the causal order, $A$ is lower triangular, i.e. all elements above the diagonal are zero. The random vector $\mathbf{X}$ also admits a "dual" representation: $\mathbf{X} = B\mathbf{e}$, where $B = (I - A)^{-1}$. To see that $B$ always exists, note that the determinant of triangular matrix is equal to the product of its diagonal entries. Since the inverse of a lower triangular matrix is lower triangular, the matrix $B$ is also lower triangular, however its diagonal elements are all equal to one. If $M = \langle \mathbf{X}, \mathbf{e}, A \rangle$, let $|M|$ be equal to the length of the vector $\mathbf{X}$. Moreover, let $\mathbf{X}(M), \mathbf{e}(M), A(M)$ and $B(M)$ be

---

[3]It is a basic fact that if $\mu << \nu$ then $\mu^n << \nu^n$.

[4]In the following $d$ refers to the total number of (potentially hidden) variables and $p \leq d$ to the number of observed variables.

$\mathbf{X}, \mathbf{e}, A$ and $(I - A)^{-1}$, respectively. The relationship between $A(M)$ and $B(M)$ will be made more perspicuous in the following.

Write $j \to_M i$ as a shorthand for $A_{ij}(M) \neq 0$. The relation $\to_M$ defines a directed acyclic graph $G(M)$ over the vertices $\{1, \ldots, |M|\}$. A causal path of length $m$ from $i$ to $j$ in $G(M)$ is a sequence of vertices $\pi = (v_1, \ldots, v_m)$ such that $v_1 = i$, $v_m = j$ and $v_i \to_M v_{i+1}$. Let $\Pi_{ij}^n(M)$ be the set of all causal paths of length $n$ from $i$ to $j$ in $G(M)$. Let $\Pi_{ij}(M)$ be the set of all causal paths from $i$ to $j$ in $G(M)$. Let $\Pi(M)$ be the set of all causal paths in $G(M)$. Write $i \rightsquigarrow_M j$ as a shorthand for $\Pi_{ij}(M) \neq \varnothing$. Write $j \circ_M i$ when $j \not\rightsquigarrow_M i$ and $j \not\leftsquigarrow_M i$. If $\pi = (v_1, \ldots, v_n)$ is a sequence of vertices in $\{1, \ldots, |M|\}$, let the **path product** $\bigtimes_M \pi$ be the product of all causal coefficients along the path $\pi$ in $G(M)$, i.e. $\bigtimes_M \pi = \prod_{i=1}^n A_{v_{i+1}, v_i}(M)$. Note that if $\pi \in \pi(M)$ iff $\bigtimes_M \pi \neq 0$.

It is easy to verify that

$$A_{ij}^k(M) = \sum_{\pi \in \Pi_{ji}^k(M)} \times_M \pi.$$

In other words, $A_{ij}^k(M)$ is the sum of all path products for paths of length $k$ from $i$ to $j$. So $A_{ij}^k(M) \neq 0$ implies $j \rightsquigarrow_M i$. The converse is not necessarily true, since non-zero path products may sum to zero. By a result of Carl Neumann's, $B(M) = \sum_{k=0}^{|M|} A^k(M)$.[5] So $B_{ij}(M) = \sum_{\pi \in \Pi_{ji}(M)} \bigtimes_M \pi$. In other words, $B_{ij}(M)$ is the sum of all path products for paths from $i$ to $j$. So $B_{ij}(M) \neq 0$ implies $j \rightsquigarrow_M i$. The converse does not necessarily hold since non-zero path products may sum to zero. We say that model $M$ is **faithful** if the total causal effect from $X_i$ to $X_j$ is nonzero if there is a causal path from $X_i$ to $X_j$. In other words: $M$ is faithful if $B_{ij}(M) \neq 0$ whenever $j \rightsquigarrow_M i$.

An acyclic linear causal model $M$ is non-Gaussian (a LiNGAM) if in addition to satisfying (1) and (2), each of the noise terms is *non-Gaussian*. Let $\mathrm{LIN}_d$ be the class of all acyclic linear causal models on $d$ variables, and let $\mathrm{LNG}_d$, $\mathrm{FLNG}_d$ respectively denote the classes of non-Gaussian models and faithful non-Gaussian models. Similarly, $\mathrm{LIN}$, $\mathrm{LNG}$ and $\mathrm{FLNG}$ respectively represent the classes of all acyclic linear causal models, all acyclic linear non-Gaussian models, and all faithful acyclic linear non-Gaussian models over some finite number of variables. It is sometimes reasonable to introduce a priori constraints on the maximum size of a coefficient in a LiNGAM model. For example, if $c$ is the number of particles in the universe, let $\mathrm{FLNG}^c$ be the set $\{M \in \mathrm{FLNG} : \max_{i,j} |B_{ij}(M)| < c\}$. Let $\mathrm{FLNG}_d^c$ be $\mathrm{FLNG}^c \cap \mathrm{FLNG}_d$.

## 4 Parsimonious Models

Let $\mathcal{O}$ be the set of all probability distributions on $\mathbb{R}^p$. We are interested in when the same vector of observed random variables could have arisen from distinct causal models. Accordingly, say that a random vector $\mathbf{O} = (O_1, \ldots, O_p) \in \mathcal{O}$ **admits** a LiNGAM model $M \in \mathrm{LNG}_d$ if there is a permutation $\alpha$ of $(1, \ldots, d)$ such that $O_i = X_{\alpha^{-1}(i)}(M)$ for $1 \leq i \leq p$. In other words: $\mathbf{O} = (O_1, \ldots, O_p)$ admits $M$ if there is a way to order the $d$ variables of $X(M)$ such that the first $p$ are identical with $O_1, \ldots, O_p$. We say that the permutation $\alpha$ **embeds $\mathbf{O}$ into $M$**. If $\alpha$ embeds $\mathbf{O}$ into $M$, then

$$\mathbf{O} = B_{\mathbf{O}}(M)\mathbf{e}_{\mathbf{O}}(M),$$

where $B_{\mathbf{O}}(M)$ is the first $p$ rows of $P_\alpha B(M) P_\alpha$, $\mathbf{e}_{\mathbf{O}}(M)$ is $P_\alpha \mathbf{e}(M)$ and $P_\alpha$ is the permutation matrix corresponding to $\alpha$. Extend the causal order over the elements of $M$ to the $O_i$ by setting $O_i \rightsquigarrow_M O_j$ if $\alpha^{-1}(i) \rightsquigarrow_M \alpha^{-1}(j)$ and $O_i \circ_M O_j$ if $\alpha^{-1}(i) \circ_M \alpha^{-1}(j)$.

Say that **$\mathbf{O}$ admits a LiNGAM model** if there is $d$ such that $\mathbf{O}$ admits $M \in \mathrm{LNG}_d$. We say that a model $M \in \mathrm{LNG}_d$ is **parsimonious** for $\mathbf{O}$ if $\mathbf{O}$ admits $M$ and $\mathbf{O}$ admits no $M'$ in $\mathrm{LNG}_f$ with $f < d$. It is immediate that if $\mathbf{O}$ admits a LiNGAM model, it admits some parsimonious LiNGAM model. For $A \in \{\mathrm{LNG}_d, \mathrm{LNG}_d^c, \mathrm{FLNG}_d, \mathrm{FLNG}_d^c\}$ Let $\mathcal{O}_A \subset \mathcal{O}$ be the set

$$\{\mathbf{O} \in \mathcal{O} : (\exists M \in A) M \text{ is parsimonious for } \mathbf{O}\}.$$

---

[5]The *spectral radius $\rho(A)$ of a square matrix $A$ is the largest absolute value of its eigenvalues. Neumann's result states that if $\rho(A) < 1$ then $(I - A)^{-1}$ exists and is equal to $\sum_{k=0}^\infty A^k$. Since the eigenvalues of a triangular matrix are exactly its diagonal entries, $\rho(B(M)) = 0$ for any acylic linear causal model $M$. By acyclicity, there are no paths longer than $|M|$, so $\sum_{k>d} A^k = 0$.*

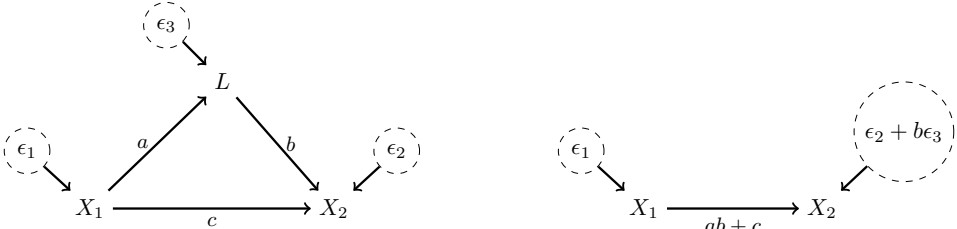

Figure 1: If $M$ is the left-hand model and $\mathbf{O} = (X_1, X_2)$, then $B_{\mathbf{O}}(M) = \begin{pmatrix} 1 & 0 & 0 \\ ab+c & 1 & b \end{pmatrix}$ and the second and third columns are proportional. The right-hand model is in fewer variables and generates the same distribution over $\mathbf{O}$.

For $A \in \{\textsc{Lng}, \textsc{Lng}^c, \textsc{Flng}, \textsc{Flng}^c\}$ Let $\mathcal{O}_{A_{\leq d}} = \cup_{j \leq d} \mathcal{O}_{A_j}$ and $\mathcal{O}_{A_{\geq d}} = \cup_{j \geq d} \mathcal{O}_{A_j}$. Let $\mathcal{O}_{A_{<d}}, \mathcal{O}_{A_{>d}}$ be defined similarly. Finally, let $\mathcal{O}_A = \mathcal{O}_{A_{\geq p}}$.

The following Lemma says that if $M$ is faithful and parsimonious for $\mathbf{O}$ then no column of $B_{\mathbf{O}}(M)$ is proportional to any other. The proof idea is expressed by Figure 1: by adjusting the edge coefficient from $X_1$ to $X_2$, it is possible to "absorb" $L$ into the noise term of $X_2$ without changing the distribution of $(X_1, X_2)$ or violating the LiNGAM model assumptions. The proof is a relatively straightforward generalization of the example, but since it is rather lengthy we relegate it to the Supplementary Material. A related result is given by Salehkaleybar et al. [2020, Theorem 11].

**Lemma 4.1.** *Suppose that $\mathbf{O}$ admits faithful $M \in \textsc{Lng}_d$ and that some column of $B_{\mathbf{O}}(M)$ is proportional to another. Then there is $M' \in \textsc{Lng}_{d-1}$ such that (i) $\mathbf{O}$ admits $M'$ and (ii) $O_i \rightsquigarrow_M O_j$ iff $O_i \rightsquigarrow_{M'} O_j$ and (iii) $M'$ is faithful.*

Lemma 4.1 raises a question about the converse: is it also the case that if $B_{\mathbf{O}}(M)$ has no two proportional columns, then $M$ is parsimonious for $\mathbf{O}$? The following Theorem from Kagan et al. [1973] allows us to answer the question in the affirmative. We will appeal to this Theorem several times in the following.

**Theorem 4.2.** *Suppose that $\mathbf{X} = A\mathbf{e} = B\mathbf{f}$, where $A$ and $B$ are $p \times r$ and $p \times s$ matrices and $\mathbf{e} = (e_1, \dots, e_r)$, $\mathbf{f} = (f_1, \dots, f_s)$ are random vectors with independent components. Suppose that no two columns of $A$ are proportional to each other. If the $i$-th column of $A$ is not proportional to any column of $B$, then $e_i$ is normally distributed.*

**Theorem 4.3.** *Suppose that faithful $M \in \textsc{Lng}_d$. Then, $M$ is parsimonious for $\mathbf{O} = (O_1, \dots, O_p)$ iff no column of $B_{\mathbf{O}}(M)$ is proportional to any other.*

*Proof of Theorem 4.3.* The left to right implication is immediate from Lemma 4.1. To prove the converse, suppose that $M$ is not parsimonious for $\mathbf{O}$. Then, $\mathbf{O}$ admits some $M' \in \textsc{Lng}_f$ with $f < d$. Moreover $\mathbf{O} = B_{\mathbf{O}}(M)\mathbf{e}_{\mathbf{O}}(M) = B_{\mathbf{O}}(M')\mathbf{e}_{\mathbf{O}}(M')$, where $B_{\mathbf{O}}(M)$ is an $p \times d$ matrix and $B_{\mathbf{O}}(M')$ is a $p \times f$ matrix. By Theorem 4.2, every column of $B_{\mathbf{O}}(M)$ is proportional to some column of $B_{\mathbf{O}}(M')$. Since the latter has fewer columns, there must be two distinct columns of $B_{\mathbf{O}}(M)$ that are proportional to the same column of $B_{\mathbf{O}}(M')$ and, therefore, to each other. $\square$

We close this section with an easy corollary of Lemma 4.1, which we will appeal to in the following.

**Corollary 4.4.** *Suppose that $\mathbf{O}$ admits faithful $M \in \textsc{Lng}$. Then there is faithful $M' \in \textsc{Lng}$ such that (i) $\mathbf{O}$ admits $M'$ (ii) $M'$ is parsimonious for $\mathbf{O}$ and (iii) $O_i \rightsquigarrow_M O_j$ iff $O_i \rightsquigarrow_{M'} O_j$.*

*Proof of Corollary 4.4.* Suppose $\mathbf{O}$ admits faithful $M \in \textsc{Lng}$ that is not parsimonious for $\mathbf{O}$. By repeated appeal to Lemma 4.1, we must eventually arrive at some $M'$ that is parsimonious for $\mathbf{O}$, either because no column of $B_{\mathbf{O}}(M)$ is proportional to any other (see Theorem 4.3), or because $M' \in \textsc{Lng}_p$ has no latent variables. $\square$

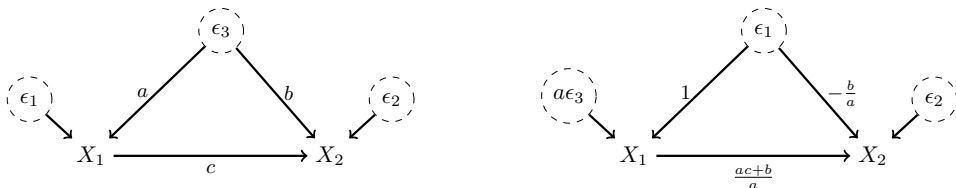

Figure 2: Note that the exogenous noise terms $\epsilon_1, \epsilon_3$ switch places. Although the left and right-hand models generate the same distribution over $(X_1, X_2)$ they disagree on the total causal effect of $X_1$ on $X_2$ whenever $b \neq 0$. When $ac = -b$, the lhs model is unfaithful and the models disagree, not only on the size of the effect, but on the presence of an edge.

## 5 Causal Identifiability

The essence of Shimizu et al. [2006] is that if a vector of $p$ observed variables admits a model in $p$ variables, that model is unique. For a simple proof, see Theorem 2.3 in Genin and Mayo-Wilson [2020].

**Theorem 5.1.** *Suppose that* $= (O_1, \ldots, O_p)$ *admits* $M, M' \in \text{LNG}_p$, *then* $M = M'$.

Unfortunately, that is no longer the case when latent variables are present. Of course, if $\mathbf{O}$ admits a LiNGAM without latents, it also admits one with latents. Figure 1 yields such an example: in the left-hand model the causal relationship is mediated by $L$ and in the right-hand model the causal relationship is unmediated. That situation is not too worrisome, since the total causal effect $X_1$ on $X_2$ is the same in both circumstances. What is more worrisome is that the vector of observed variables $\mathbf{O}$ may admit two LiNGAM models that *differ* on the effects of interventions. An example, due to Salehkaleybar et al. [2020], is given in Figure 2. The good news is that if a set of observed variables admits two faithful LiNGAM models, the models must agree on the ancestor relationship between them. Although this is shown already by Salehkaleybar et al. [2020, Lemma 5], the following is a simple proof that does not rely on facts about independent component analysis.

**Theorem 5.2.** *Suppose that* $\mathbf{O} = (O_1, \ldots, O_p)$ *admits faithful* $M, M'$. *Then* $O_i \rightsquigarrow_M O_j$ *iff* $O_i \rightsquigarrow_{M'} O_j$

*Proof of Theorem 5.2.* By Corollary 4.4 there are faithful LiNGAMs $F, F'$ such that 1. $\mathbf{O}$ admits $F, F'$; 2. $O_i \rightsquigarrow_M O_j$ iff $O_i \rightsquigarrow_F O_j$; 3. $O_i \rightsquigarrow_{M'} O_j$ iff $O_i \rightsquigarrow_{F'} O_j$ and 4. $B_{\mathbf{O}}(F)$ and $B_{\mathbf{O}}(F')$ both have pairwise linearly independent columns. By (1) and (2), it suffices to prove that $O_i \rightsquigarrow_F O_j$ iff $O_i \rightsquigarrow_{F'} O_j$. But since the situation is symmetrical, it suffices to prove that $O_i \rightsquigarrow_F O_j$ only if $O_i \rightsquigarrow_{F'} O_j$.

Suppose for a contradiction that $O_i \rightsquigarrow_F O_j$ but $O_i \not\rightsquigarrow_{F'} O_j$. Let $\alpha$ be a permutation embedding $\mathbf{O}$ in $F$. Let $B, C$ be $B_{\mathbf{O}}(F), B_{\mathbf{O}}(F')$, respectively. Let $\mathbf{e}, \mathbf{f}$ be $\mathbf{e}_{\mathbf{O}}(F), \mathbf{e}_{\mathbf{O}}(F')$, respectively. Then

$$\mathbf{O} = B\mathbf{e} = C\mathbf{f}.$$

Since $O_i \not\rightsquigarrow_{F'} O_j$, $C_{ji} = 0$. Moreover, $C_{ii} = 1$ By faithfulness of $F$, $O_i \rightsquigarrow_F O_j$ implies that $B_{ji} \neq 0$. By Theorem 4.2, there must be a column $k \neq i$ and real number $a \neq 0$ such that $B_{ik} = aC_{ii} \neq 0$ but $B_{jk} = aC_{ji} = 0$. Since $B_{ik} \neq 0$, it follows that $\alpha^{-1}(k) \rightsquigarrow_F \alpha^{-1}(i)$. Since $O_i \rightsquigarrow_F O_j$ by assumption, it follows that $\alpha^{-1}(i) \rightsquigarrow_F \alpha^{-1}(j)$. By transitivity of $\rightsquigarrow_F$, $\alpha^{-1}(k) \rightsquigarrow_F \alpha^{-1}(j)$. However, $B_{jk} = 0$. So $F$ is unfaithful. Contradiction. $\square$

For $A \in \{\text{LNG}, \text{LNG}^c, \text{LNG}_d^c, \text{FLNG}, \text{FLNG}^c, \text{FLNG}_d^c\}$, let $\mathcal{O}_A^{i \to j}$ be the set

$$\{\mathbf{O} \in \mathcal{O}_A : (\exists M \in A) \ \mathbf{O} \text{ admits } M \text{ and } O_i \to_M O_j\}.$$

Define $\mathcal{O}_A^{i \rightsquigarrow j}, \mathcal{O}_A^{i \circ j}$ analogously, replacing $\to_M$ with $\rightsquigarrow_M$ and $\circ_M$, respectively. In light of Theorem 5.2, $\mathcal{O}_{\text{FLNG}}^{i \rightsquigarrow j}, \mathcal{O}_{\text{FLNG}}^{i \leftarrow j}$ and $\mathcal{O}_{\text{FLNG}}^{i \circ j}$ are disjoint.

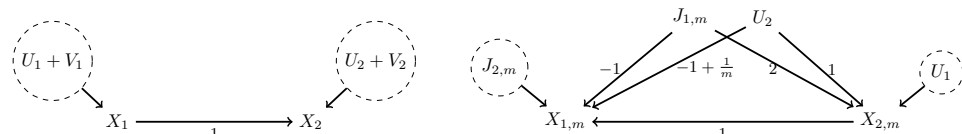

Figure 3: The $(X_{1,m}, X_{2,m})$, which lie in $\mathcal{O}^{1 \leftarrow 2}_{\text{FLNG}^c_4}$, converge in probability to $(X_1, X_2)$, lying in $\mathcal{O}^{1 \rightarrow 2}_{\text{FLNG}^c_2}$. Note that although error terms approach Gaussianity and the model approaches unfaithfulness, no term in the sequence is unfaithful and no noise term is Gaussian. For definitions of error and exogenous terms, see the proof of Lemma 6.2.

## 6 The Topology of Latent Confounding

When $\mathbf{O} = (O_1, \ldots, O_p)$ admits a LiNGAM model without latents, Genin and Mayo-Wilson [2020, Theorem 4.1] prove that orientation hypotheses are topologically well-separated:

**Lemma 6.1.** $\mathcal{O}^{i \rightarrow j}_{\text{LNG}^c_p}, \mathcal{O}^{i \leftarrow j}_{\text{LNG}^c_p}$ are open and $\mathcal{O}^{i \circ j}_{\text{LNG}^c_p}$ is closed in the weak topology on $\mathcal{O}_{\text{LNG}^c_p}$.

The situation changes when we allow for latent variables.

**Lemma 6.2.** $\mathcal{O}^{i \rightarrow j}_{\text{FLNG}^c_p}$ is not disjoint from $\text{cl}(\mathcal{O}^{i \leftarrow j}_{\text{FLNG}^c_{p+2}})$ in the weak topology on $\mathcal{O}_{\text{FLNG}^c}$. Moreover, there are distributions in the intersection that are absolutely continuous wrt Lebesgue measure.

*Proof of Lemma 6.2.* Let $p = 2$. Let $U_1, U_2, W_1, W_2, Z_1, Z_2$ be mutually independent, absolutely continuous random variables. Suppose that all variables except $Z_1, Z_2$ are non-Gaussian. Let $V_1 = Z_1 + Z_2$ and let $V_2 = Z_1 - Z_2$. By the Lukacs-King theorem, $V_1, V_2$ are independent. Let $\mathbf{X} = (X_1, X_2) = (U_1 + V_1, U_1 + U_2 + 2Z_1)$. By reference to the lhs model in Figure 3, it is clear that $\mathbf{X} \in \mathcal{O}_{\text{FLNG}^c_2}$. Moreover, $\mathbf{X}$ is absolutely continuous wrt Lebesgue measure on $\mathbb{R}^2$.

For $m > 2$, let $X_{1,m} = U_1 + V_1 + \frac{1}{m}(W_1 + W_2 + U_2)$ and $X_{2,m} = U_1 + U_2 + 2Z_1 + \frac{2}{m}W_1$. Let $\mathbf{X}_m = (X_{1,m}, X_{2,m})$. It is clear that the $\mathbf{X}_m$ converges in probability, and therefore in distribution, to $\mathbf{X}$. It remains to show that the $\mathbf{X}_m$ lie in $\mathcal{O}_{\text{FLNG}^c}$, which we do by reference to the rhs model in Figure 3. Let $J_{1,m} = Z_1 + \frac{1}{m}W_1$ and $J_{2,m} = Z_2 + \frac{1}{m}W_2$. Then $J_{1,m}, J_{2,m}, U_1, U_2$ are independent and non-Gaussian. Let $\mathbf{e}^T_m = (J_{2,m}, U_1, J_{1,m}, U_2)$. Let $A_m = \begin{pmatrix} 0 & 1 & -1 & -1+\frac{1}{m} \\ 0 & 0 & 2 & 1 \\ 0 & 0 & 0 & 0 \\ 0 & 0 & 0 & 0 \end{pmatrix}$ and

$B_m = \begin{pmatrix} 1 & 1 & 1 & \frac{1}{m} \\ 0 & 1 & 2 & 1 \\ 0 & 0 & 1 & 0 \\ 0 & 0 & 0 & 1 \end{pmatrix}$. It is easy to check that $B_m = (I - A_m)^{-1}$. Let $M_m = \langle B_m \mathbf{e}_m, A_m, \mathbf{e}_m \rangle$.

By inspection of $B_m$, $M_m$ is faithful. Since the entries of $A_m$ are smaller than $c$, $M_m \in \text{FLNG}^c_4$. Letting $C_m$ be the first two rows of $B_m$, it is easy to verify that $(X_{1,m}, X_{2,m})^T = C_m \mathbf{e}_m$. By Theorem 4.3, since, for $m > 2$, no column of $C_m$ is proportional to any other, $M_m$ is parsimonious for $\mathbf{X}_m$. Therefore, $\mathbf{X}_m \in \mathcal{O}_{\text{FLNG}^c_4}$. $\square$

In the following, we will appeal extensively to the following Lemma, given by Kagan et al. [1973].

**Lemma 6.3.** *Suppose the $k$-dimensional random vectors $\mathbf{e}_n$ have independent components. Consider the sequence of $p$-dimensional random vectors $\mathbf{X}_n = B\mathbf{e}_n$, where $B$ is a $p \times k$ matrix. If the $\mathbf{X}_n$ converge in distribution to $\mathbf{X}$, then $X = B\mathbf{e}$, where $\mathbf{e}$ is a $k$-dimensional random vector with independent components.*

The following is a straightforward Corollary of Lemma 6.3.

**Corollary 6.4.** *Suppose the $k$-dimensional random vectors $\mathbf{e}_n$ have independent components. Consider a sequence of $p$-dimensional random vectors $\mathbf{X}_n = B_n \mathbf{e}_n$, where the $B_n$ are $p \times k$ matrices and $B_n \rightarrow B$. If the $\mathbf{X}_n$ converge in distribution to $\mathbf{X}$, then $X = B\mathbf{e}$, where $\mathbf{e}$ is a $k$-dimensional random vector with independent components.*

*Proof of Corollary 6.4.* It is a standard fact that if $|X_n - Y_n|$ converge in probability to $0$ and the $X_n$ converge in distribution to $X$, then the $Y_n$ also converge in distribution to $X$. Clearly, $|B_n\mathbf{e}_n - B\mathbf{e}_n|$ converge in probability to $0$. By assumption, the $B_n\mathbf{e}_n$ converge in distribution to $\mathbf{X}$. It follows that $B\mathbf{e}_n$ converge in distribution to $\mathbf{X}$. By Lemma 6.3, $X = B\mathbf{e}$, where $\mathbf{e}$ is a $k$-dimensional random vector with independent components. $\qquad\square$

**Theorem 6.5.** *For $d \geq p$, $\mathcal{O}_{\mathrm{FLNG}^c_{>d}}$ is open in the weak topology on $\mathcal{O}_{\mathrm{FLNG}^c}$.*

*Proof of Theorem 6.5.* Let $e \leq d < f$ Suppose for a contradiction that the $\mathbf{O}_n \in \mathrm{FLNG}^c_e$ converge in distribution to $\mathbf{O} \in \mathrm{FLNG}^c_f$. Let $M \in \mathrm{FLNG}^c_f$ be parsimonious for $\mathbf{O}$ and $M_n \in \mathrm{FLNG}^c_e$ be parsimonious for $\mathbf{O}_n$. Let $B_n = B_{\mathbf{O}_n}(M_n)$ and $A = B_{\mathbf{O}}(M)$. Let $\mathbf{e}_n = \mathbf{e}_{\mathbf{O}_n}(M_n)$ and $\mathbf{e} = \mathbf{e}_{\mathbf{O}}(M)$. While $B$ is a $p \times f$ matrix, each of the $B_n$ are $p \times e$ matrices. By the Bolzano-Weierstrass theorem, since the $B_n$ are uniformly bounded, there is a $p \times e$ matrix $B$ and a convergent subsequence $B_{n_m} \to B$. By assumption, $B_{n_m}\mathbf{e}_{n_m}$ converge in distribution to $\mathbf{O}$. By Corollary 6.4, $\mathbf{O} = B\mathbf{f}$ where $\mathbf{f}$ is a vector of independent components. Therefore $\mathbf{O} = A\mathbf{e} = B\mathbf{f}$. By 4.2 every column of $A$ must be proportional to some column of $B$. Since $A$ has strictly more columns than $B$, two columns of $A$ must be proportional to the same column of $B$ and, therefore, to each other. But then, by Theorem 4.3, $M$ is not parsimonious for $\mathbf{O}$. Contradiction.

We have shown that for every $\mathbf{O} \in \mathrm{FLNG}^c_f$, there is an open set separating $\mathbf{O}$ from $\mathrm{FLNG}^c_e$. Since $e < f$ was taken to be arbitrary, there is such an open set $E_g$ separating $\mathbf{O}$ from each $\mathrm{FLNG}^c_g$ with $p \leq g < f$. Since there are only finitely many of the $E_g$, the intersection of the $E_g$ is open and separates $\mathbf{O}$ from $\mathcal{O}_{\mathrm{FLNG}^c_{\leq d}}$. That shows that for every $\mathbf{O} \in \mathcal{O}_{\mathrm{FLNG}^c_{>d}}$, there is an open set $E_{\mathbf{O}}$ separating $\mathbf{O}$ from $\mathcal{O}_{\mathrm{FLNG}^c_{\leq d}}$. Therefore, $\mathcal{O}_{\mathrm{FLNG}^c_{>d}} = \cup_{\mathbf{O} \in \mathcal{O}_{\mathrm{FLNG}^c_{>d}}} E_{\mathbf{O}}$ is a union of open sets and, therefore, open in $\mathcal{O}_{\mathrm{FLNG}^c}$. $\qquad\square$

As a special case, Theorem 6.5 entails that $\mathcal{O}_{\mathrm{FLNG}^c_{>p}}$ is open. By Genin and Kelly [2017, Theorem 4.1], this means that it is statistically verifiable whether an unobserved confounder must be introduced in order to accommodate the distribution of $\mathbf{O}$, at least when all distribution are assumed to be absolutely continuous wrt Lebesgue measure. As expected, the hypothesis of un-confoundedness is statistically testable. On the other hand, the precise hypothesis $\mathrm{FLNG}^c_d$ is neither statistically verifiable nor refutable, even under the background assumption that the distribution $\mathbf{O}$ was generated by some model in $\mathrm{FLNG}^c$. To see this, note that for $d > p$, $\mathrm{FLNG}^c_d$ is neither open nor closed, since more parsimonious models can approximate simpler models. Therefore, it is properly locally closed. Although it is neither verifiable nor decidable, it is decidable in the limit by Theorem 2.1.[6] We shall see that the same is true for the hypothesis of orientation $\mathcal{O}^{i \rightsquigarrow j}_{\mathrm{FLNG}^c}$. The following shows that if we knew exactly how many latent variables were necessary to accommodate the observed distribution, orientation hypotheses would be topologically well-separated.

**Theorem 6.6.** *For $d \geq p$, $\mathcal{O}^{i \rightsquigarrow j}_{\mathrm{FLNG}^c_d}, \mathcal{O}^{i \leftsquigarrow j}_{\mathrm{FLNG}^c_d}$ are open and $\mathcal{O}^{i \circ j}_{\mathrm{FLNG}^c_d}$ is closed in the weak topology on $\mathcal{O}_{\mathrm{FLNG}^c_d}$.*

*Proof of Theorem 6.6.* Suppose for a contradiction that the $\mathbf{O}_n \in \mathcal{O}^{i \not\rightsquigarrow j}_{\mathrm{FLNG}^c_d}$ converge in distribution to $\mathbf{O} \in \mathcal{O}^{i \rightsquigarrow j}_{\mathrm{FLNG}^c_d}$. Let $M \in \mathrm{FLNG}^c_d$ be parsimonious for $\mathbf{O}$ and $M_n \in \mathrm{FLNG}^c_d$ be parsimonious for $\mathbf{O}_n$. Let $B_n = B_{\mathbf{O}_n}(M_n)$ and $A = B_{\mathbf{O}}(M)$. Let $\mathbf{e}_n = \mathbf{e}_{\mathbf{O}_n}(M_n)$ and $\mathbf{e} = \mathbf{e}_{\mathbf{O}}(M)$. By the Bolzano-Weierstrass theorem, since the $B_n$ are uniformly bounded, there is a $p \times e$ matrix $B$ and a convergent subsequence $B_{n_m} \to B$. By assumption, $B_{n_m}\mathbf{e}_{n_m}$ converge in distribution to $\mathbf{O}$. By Corollary 6.4, $\mathbf{O} = B\mathbf{f}$ where $\mathbf{f}$ is a vector of independent components. Therefore $\mathbf{O} = A\mathbf{e} = B\mathbf{f}$. Since $(B_n)_{ji} = 0$ for all $n$, $B_{ji} = 0$. Moreover, $B_{ii} = 1$. Since $A$ and $B$ have equal dimensions, by Theorem 4.2 there must be a column $k$ such that $A_{jk} = 0$ and $A_{ik} \neq 0$. But then $O_k \rightsquigarrow_M O_i$ and $O_i \rightsquigarrow_M O_j$ and, therefore, $O_k \rightsquigarrow_M O_j$. But since $A_{jk} = 0$, $M$ must be unfaithful. Contradiction. We have shown that $\mathcal{O}^{i \rightsquigarrow j}_{\mathrm{FLNG}^c_d}$ is open in the weak topology on $\mathcal{O}_{\mathrm{FLNG}^c_d}$. Since the situation is symmetrical, $\mathcal{O}^{i \leftsquigarrow j}_{\mathrm{FLNG}^c_d}$ is also open. Since $\mathcal{O}^{i \circ j}_{\mathrm{FLNG}^c_d}$ is the complement of $\mathcal{O}^{i \rightsquigarrow j}_{\mathrm{FLNG}^c_d} \cup \mathcal{O}^{i \leftsquigarrow j}_{\mathrm{FLNG}^c_d}$, it is closed in $\mathcal{O}_{\mathrm{FLNG}^c_d}$. $\qquad\square$

---

[6]Recall that, in metrizable spaces such as the weak topology, every locally closed set is a countable union of closed sets.

**Corollary 6.7.** $\mathcal{O}^{i \rightsquigarrow j}_{\mathrm{FLNG}^c}, \mathcal{O}^{i \leftsquigarrow j}_{\mathrm{FLNG}^c}, \mathcal{O}^{i \circ j}_{\mathrm{FLNG}^c}$ *are disjoint countable unions of sets closed in* $\mathcal{O}_{\mathrm{FLNG}^c}$.

*Proof of Corollary 6.7.* In general if $A$ is open/closed in a subspace, it is the intersection of an open/closed set with the subspace. By Theorem 6.5, $\mathcal{O}_{\mathrm{FLNG}^c_d}$ is locally closed in $\mathcal{O}_{\mathrm{FLNG}^c}$. By Theorem 6.6, $\mathcal{O}^{i \rightsquigarrow j}_{\mathrm{FLNG}^c_d}, \mathcal{O}^{i \leftsquigarrow j}_{\mathrm{FLNG}^c_d}, \mathcal{O}^{i \circ j}_{\mathrm{FLNG}^c_d}$ are either open or closed in $\mathcal{O}_{\mathrm{FLNG}^c_d}$. Therefore, $\mathcal{O}^{i \rightsquigarrow j}_{\mathrm{FLNG}^c_d}, \mathcal{O}^{i \leftsquigarrow j}_{\mathrm{FLNG}^c_d}, \mathcal{O}^{i \circ j}_{\mathrm{FLNG}^c_d}$ are locally closed in $\mathcal{O}_{\mathrm{FLNG}^c}$. It follows that each of $\mathcal{O}^{i \rightsquigarrow j}_{\mathrm{FLNG}^c}, \mathcal{O}^{i \leftsquigarrow j}_{\mathrm{FLNG}^c}, \mathcal{O}^{i \circ j}_{\mathrm{FLNG}^c}$ is a countable union of locally closed sets. In a metrizable space such as the weak topology, each open set, and therefore each locally closed set, is a countable union of closed sets. Therefore, each of $\mathcal{O}^{i \rightsquigarrow j}_{\mathrm{FLNG}^c}, \mathcal{O}^{i \leftsquigarrow j}_{\mathrm{FLNG}^c}, \mathcal{O}^{i \circ j}_{\mathrm{FLNG}^c}$ is a countable union of closed sets. They are disjoint by Theorem 5.2. □

## 7 Main Result and Discussion

We are now in position to state and prove the main results. Let $\mathcal{M}$ be the set of all pairs $\langle M, \alpha \rangle$, where $M \in \mathrm{FLNG}^c$ and $\alpha$ is a permutation of $\{1, \dots, |M|\}$. Let $P(\langle M, \alpha \rangle) = (\mathbf{X}_{\alpha^{-1}(1)}(M), \dots, \mathbf{X}_{\alpha^{-1}(p)}(M))$. Let $\mathcal{M}^{i \rightsquigarrow j} = \{\langle M, \alpha \rangle \in \mathcal{M} : \alpha^{-1}(i) \rightsquigarrow_M \alpha^{-1}(j)\}$ and $\mathcal{M}^{i \circ j} = \{\langle M, \alpha \rangle \in \mathcal{M} : \alpha^{-1}(i) \circ_M \alpha^{-1}(j)\}$.

**Theorem 7.1.** *The question* $\mathfrak{Q} = \{\mathcal{M}^{i \rightsquigarrow j}, \mathcal{M}^{i \circ j}, \mathcal{M}^{i \leftsquigarrow j}\}$ *is identified and decidable in the limit, but not statistically decidable.*

*Proof of Theorem 7.1.* It is immediate from defintions that $P(\mathcal{M}^{i \rightsquigarrow j}) = \mathcal{O}^{i \rightsquigarrow j}_{\mathrm{FLNG}^c}$, $P(\mathcal{M}^{i \leftsquigarrow j}) = \mathcal{O}^{i \leftsquigarrow j}_{\mathrm{FLNG}^c}$ and $P(\mathcal{M}^{i \circ j}) = \mathcal{O}^{i \circ j}_{\mathrm{FLNG}^c}$. Since $\mathcal{O}^{i \rightsquigarrow j}_{\mathrm{FLNG}^c}, \mathcal{O}^{i \leftsquigarrow j}_{\mathrm{FLNG}^c}, \mathcal{O}^{i \circ j}_{\mathrm{FLNG}^c}$ are disjoint by Theorem 5.2, the question is identified. Since they are each countable unions of closed sets (by Corollary 6.7), Theorem 2.1 implies that $\mathfrak{Q}$ is decidable in the limit. By Theorem 2.2 and Lemma 6.2, $\mathfrak{Q}$ is not decidable. □

Theorem 7.1 shows that learning causal orientation in faithful, but potentially confounded, LiNGAM models is a difficult problem. Not so difficult that it is impossible to construct consistent methods, but difficult enough that no consistent method can guarantee a finite-sample bound on the probability of orientation errors. In view of the positive results given by Genin and Mayo-Wilson [2020] in the unconfounded setting, this negative result is something of a disappointment. However, the example in Figure 3 suggests several different adjustments to the LiNGAM framework that may recover decidability. The first is the well-trodden path of strong faithfulness. As $m$ grows, the direct path from $U_2$ to $X_{1,m}$ comes closer and closer to cancelling the path via $X_{2,m}$. Strengthening faithfulness would preclude this possibility. But faithfulness is already a controversial assumption and strengthenings would do nothing to appease its critics. Moreover, Uhler et al. [2013] show that strong versions of faithfullness can rule out a topologically large set of models. The second path of escape is to strengthen the assumption of non-Gaussianity. As $m$ grows, $J_{1,m}$ and $J_{2,m}$ converge to the Gaussians $Z_1$ and $Z_2$. Assuming that noise terms are bounded away from Gaussianity would preclude this possibility. The third possibility is to require that no error term have a *Gaussian component*. A random variable $X$ has a Gaussian component if it can be expressed as the sum $Y + Z$ where $Y, Z$ are independent and $Z$ is Gaussian. It is clear that the error terms in Figure 3 violate this condition — indeed properties of the Gaussian are essential to ensuring that $V_1$ and $V_2$, and therefore $U_1 + V_1$ and $U_2 + V_2$ are independent. In light of uniqueness results by Kagan et al. [1973] (see particularly their Theorem 10.3.6) it is likely that requiring that noise terms have no Gaussian component would recover decidability. Moreover, the assumption of no Gaussian components does not strike this author as significantly less plausible than the assumption of non-Gaussianity.

In recent years, the field of causal discovery has produced many exciting new identifiability results under a variety of modeling assumptions. But demonstrating identifiability proves only that a problem is not completely hopeless — it is only the first step in understanding how difficult a problem is. Success notions intermediate between uniform decidability and decidability in the limit can help structure the search for "Goldilocks" modeling assumptions: neither so weak as to preclude short-run bounds on error nor so strong as to secure uniform convergence. Moreover, it is the hope of this author that the topological methods exhibited here prove useful in these future investigations.

## Acknowledgments and Disclosure of Funding

Funded by the Deutsche Forschungsgemeinschaft (DFG, German Research Foundation) under Germany's Excellence Strategy – EXC number 2064/1 – Project number 390727645. I am grateful to Conor Mayo-Wilson and the anonymous reviewers for their careful reading and helpful comments and suggestions.

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
