# A Appendix

*Proof of Lemma 4.1.* Suppose that $\alpha$ embeds $\mathbf{O} = (O_1, \ldots, O_p)$ in $M \in \mathrm{LNG}_d$ and that $D = B_{\mathbf{O}}(M)$. Suppose that $D_{[,v]} = a D_{[,u]}$. First, we show that at least one of $u, v$ must be strictly greater than $p$. Suppose for a contradiction that $u, v \leq p$. Since $D$ has ones everywhere on the diagonal of the principal $p \times p$ submatrix, $D_{uu} = D_{vv} = 1$. By assumption, we have that $D_{uv} = a > 0$ and $D_{vu} = 1/a > 0$. But then $O_u \leadsto_M O_v$ and $O_v \leadsto_M O_u$, which contradicts acyclicity. Therefore, without loss of generality, we suppose that $v > p$.

Let $y, z = \alpha^{-1}(u), \alpha^{-1}(v)$. Let $A = A(M)$ and $B = B(M)$. Let $\beta$ be the permutation of $\{1, \ldots, d\}$ sending $i \mapsto i$ for $i < z$ and $i \mapsto i - 1$ for $i > z$. Define the $d - 1 \times d - 1$ matrix $A'$ in the following way:

$$A'_{ij} = A_{\beta^{-1}(i,j)} + A_{z,\beta^{-1}(j)} A_{\beta^{-1}(i),z}.$$

Since $A$ has zeros on the diagonal and, by acyclicity, one of $A_{z,\beta^{-1}(i)}, A_{\beta^{-1}(i),z}$ must be zero, $A'$ has zeros on the diagonal. We show that $A'$ is lower triangular, i.e. that $A'_{ij} = 0$ whenever $j > i$. There are three cases to consider: (1) $i < z, j < z$; (2) $i < z, j \geq z$ and (3) $i \geq z, j \geq z$. In the first case $A'_{ij} = A_{ij} + A_{zj} A_{iz}$. Since $A$ is lower triangular, $A_{ij} = A_{iz} = 0$. In the second case, $A'_{ij} = A_{i,j+1} + A_{z,j+1} A_{iz}$. Since $A$ is lower triangular, $A_{i,j+1} = A_{iz} = 0$. In the final case, $A'_{ij} = A_{i+1,j+1} + A_{z,j+1} A_{i+1,z}$. Since $A$ is lower triangular, $A_{i+1,j+1} = A_{z,j+1} = 0$.

Since $I - A'$ is lower triangular and its diagonal entries are all equal to one, the inverse matrix $B' = (I - A')^{-1}$ exists. We argue that $B'_{ij} = B_{\beta^{-1}(i,j)}$. Let $\Pi_{ij}$ be the set of all paths from $i$ to $j$ over the vertices $\{1, \ldots, d\}$. Let $\Pi_{ikj}$ be the set of all paths from $i$ to $j$ over the vertices $\{1, \ldots, d\}$ passing through $k$ and let $\Pi_{i\not k j} = \Pi_{ij} \setminus \Pi_{ikj}$. Let $\Pi'_{ij}$ bet the set of all paths from $i$ to $j$ over the vertices $\{1, \ldots, d - 1\}$. From our previous observation, we have that

$$B'_{ji} = \sum_{\pi \in \Pi'_{ij}} \prod_{i=1}^{|\pi|} A'_{\pi_{i+1}, \pi_i} \tag{1}$$

$$= \sum_{\pi \in \Pi'_{ij}} \prod_{i=1}^{|\pi|} \left( A_{\beta^{-1}(\pi_{i+1}, \pi_i)} + A_{z, \beta^{-1}(\pi_i)} A_{\beta^{-1}(\pi_{i+1}), z} \right) \tag{2}$$

$$= \sum_{\pi \in \Pi_{\beta^{-1}(i) \not z \beta^{-1}(j)}} \prod_{i=1}^{|\pi|} \left( A_{\pi_{i+1}, \pi_i} + A_{z, \pi_i} A_{\pi_{i+1}, z} \right). \tag{3}$$

Note that for any $\pi \in \Pi_{ij}$ there can be at most one $\pi_i$ such that

$$A_{z, \pi_i} A_{\pi_{i+1}, z} \neq 0.$$

If this were not the case, there would be causal paths in $G(M)$ passing through $z$ twice, contradicting acyclicity. Let $\pi_{i*}$ be the unique such $\pi_i$, if it exists, and let $\pi_{i*} = \pi_1$, otherwise. Then:

$$B'_{ji} = \sum_{\pi \in \Pi_{\beta^{-1}(i) \not z \beta^{-1}(j)}} \prod_{i=1}^{|\pi|} (A_{\pi_{i+1}, \pi_i}) + A_{z, \pi_{i*}} A_{\pi_{i*+1}, z} \prod_{i \neq i*} (A_{\pi_{i+1}, \pi_i}) \tag{4}$$

$$= \sum_{\pi \in \Pi_{\beta^{-1}(i) \not z \beta^{-1}(j)}} \prod_{i=1}^{|\pi|} (A_{\pi_{i+1}, \pi_i}) + \sum_{\pi \in \Pi_{\beta^{-1}(i) z \beta^{-1}(j)}} \prod_{i=1}^{|\pi|} (A_{\pi_{i+1}, \pi_i}) \tag{5}$$

$$= \sum_{\pi \in \Pi_{\beta^{-1}(i) \not z \beta^{-1}(j)}} \times_M \pi + \sum_{\pi \in \Pi_{\beta^{-1}(i) z \beta^{-1}(j)}} \times_M \pi \tag{6}$$

$$= \sum_{\pi \in \Pi_{\beta^{-1}(i,j)}} \times_M \pi \tag{7}$$

$$= B_{\beta^{-1}(j,i)}. \tag{8}$$

Let the $d-1$ element column vector $\mathbf{e}'$ be just like the first $d-1$ rows of $P_\beta \mathbf{e}(M)$ except $\mathbf{e}'_{\beta(y)} = \mathbf{e}_y + a\mathbf{e}_z$. Since the sum of independent non-Gaussian variables is non-Gaussian, each element of $\mathbf{e}'$ is not Gaussian. Moreover, since functions of independent random variables are independent, the $\mathbf{e}'_i$ are mutually independent.

Let $\mathbf{X}' = B'\mathbf{e}'$ Since $B'$ is lower triangular with ones on the diagonal and $\mathbf{e}'$ is a vector of mutually independent, non-Gaussian random variables, we have that $M' = \langle \mathbf{X}', \mathbf{e}', A' \rangle$ is in $\mathrm{LNG}_{d-1}$.

We are now in a position to prove part (i) of the theorem. We claim that $\beta^{-1} \circ \alpha$ embeds $\mathbf{O}$ into $M'$. In other words, we claim that $O_i = \mathbf{X}_{\beta(\alpha^{-1}(i))}(M')$ :

$$
\begin{aligned}
\mathbf{X}_{\beta(\alpha^{-1}(i))}(M') &= \sum_{j=1}^{d-1} B'_{\beta(\alpha^{-1}(i)),j}\mathbf{e}'_j \\
&= B'_{\beta(\alpha^{-1}(i)),\beta(y)}\mathbf{e}'_{\beta(y)} + \sum_{j<z, j\neq\beta(y)} B'_{\beta(\alpha^{-1}(i)),j}\mathbf{e}'_j + \sum_{j\geq z, j\neq\beta(y)} B'_{\beta(\alpha^{-1}(i)),j}\mathbf{e}'_j \\
&= B_{\alpha^{-1}(i),y}(\mathbf{e}_y + a\mathbf{e}_z) + \sum_{j<z, j\neq\beta(y)} B_{\alpha^{-1}(i),\beta^{-1}(j)}\mathbf{e}'_j + \sum_{j\geq z, j\neq\beta(y)} B_{\alpha^{-1}(i),\beta^{-1}(j)}\mathbf{e}'_j \\
&= B_{\alpha^{-1}(i),y}(\mathbf{e}_y + a\mathbf{e}_z) + \sum_{j<z, j\neq y} B_{\alpha^{-1}(i),j}\mathbf{e}_j + \sum_{j\geq z, j\neq y} B_{\alpha^{-1}(i),j+1}\mathbf{e}_{j+1} \\
&= B_{\alpha^{-1}(i),y}(\mathbf{e}_y + a\mathbf{e}_z) + \sum_{j<z, j\neq y} B_{\alpha^{-1}(i),j}\mathbf{e}_j + \sum_{j>z, j\neq y} B_{\alpha^{-1}(i),j}\mathbf{e}_j \\
&= aB_{\alpha^{-1}(i),y}\mathbf{e}_z + \sum_{j\neq z} B_{\alpha^{-1}(i),j}\mathbf{e}_j \\
&= \mathbf{X}_{\alpha^{-1}(i)}(M) \\
&= O_i.
\end{aligned}
$$

The third and fourth equalities follows from the fact that $B'_{ij} = B_{\beta^{-1}(i,j)}$ and that $\beta^{-1}(i) = i$ when $i < z$ and $\beta^{-1}(i) = i+1$ when $i \geq z$. The penultimate equality follows from the fact that $aB_{\alpha^{-1}(i),y}\mathbf{e}_z = aB_{\alpha^{-1}(i),\alpha^{-i}(u)}\mathbf{e}_z = aD_{i,u}\mathbf{e}_z = D_{i,v}\mathbf{e}_z = B_{\alpha^{-1}(i),\alpha^{-i}(v)}\mathbf{e}_z = B_{\alpha^{-1}(i),z}\mathbf{e}_z$. The final equality follows from the fact that $\alpha$ embeds $\mathbf{O}$ in $M$.

We now prove (ii). Suppose that $O_i \rightsquigarrow_M O_j$. Since $M$ is faithful, $B_{\alpha^{-1}(j,i)} \neq 0$ and therefore $B'_{\beta(\alpha^{-1}(j,i))} \neq 0$. That entails that $\beta(\alpha^{-1}(i)) \rightsquigarrow_{M'} \beta(\alpha^{-1}(j))$ and, since $\beta^{-1} \circ \alpha$ embeds $\mathbf{O}$ in $M'$, $O_i \rightsquigarrow_{M'} O_j$. For the converse it suffices to show that $i \rightarrow_{M'} j$ entails $\beta^{-1}(i) \rightsquigarrow_M \beta^{-1}(j)$. Suppose the antecedent holds. Then $A'_{ji} \neq 0$. Therefore, either $A_{\beta^{-1}(j,i)} \neq 0$ or $A_{z,\beta^{-1}(i)}A_{\beta^{-1}(j),z} \neq 0$. In either case, $\beta^{-1}(i) \rightsquigarrow_M \beta^{-1}(j)$.

It remains to prove (iii). Suppose that $i \rightsquigarrow_{M'} j$. Then, by (ii), $\beta^{-1}(i) \rightsquigarrow \beta^{-1}(j)$. Since $M$ is faithful, $B_{\beta^{-1}(j,i)} \neq 0$. But since $B_{\beta^{-1}(j,i)} = B'_{j,i}$, $B'_{j,i} \neq 0$, as required. $\qquad\square$