# OpenReview forum: "Statistical Undecidability in Linear, Non-Gaussian Causal Models in the Presence of Latent Confounders"
_NeurIPS.cc/2021/Conference — NeurIPS 2021 Poster_

### Official Review · Reviewer_YVg9 · 2021-07-11

**Rating:** 8
**Confidence:** 4

**Summary:**

This paper is about statistical causal inference. More specifically, this paper is about the causal discovery that infers causal graphs. Many methods focus on investigating the identifiability of various models for causal discovery and proposing estimation methods for the identifiable models. This paper points out that the identifiability does not necessarily imply that desirably consistent estimations methods exist.

There are many types of consistency, including uniform consistency and point-wise consistency. This paper investigates which consistency can be achieved for linear non-Gaussian acyclic models with hidden common causes, confounded LiNGAM (Hoyer et al., 2008). This analysis is based on a consistency concept named statistical decidability proposed in Genin and Mayo-Wilson (2020).

Their main result is summarized in Table 1. Genin and Mayo-Wilson (2020) showed that LiNGAM with latent confounders (Shimizu et al., 2006) is (statistically) decidable with/without the faithfulness assumption. Faithfulness is a common assumption in causal discovery (Spirtes et al., 1993). The causal ordering of variables, ancestral relationships, of confounded LiNGAM has been shown in identifiable if faithfulness is assumed (Hoyer et al., 2008; Salehkaleybar et al., 2020). If faithfulness is not assumed, it is not decidable. The definitions of decidability are on pages 2 and 3. Theorem 1.1 and 1.2 say how one could check whether such consistency hold. Their very main result is Theorem 6.1 in the final section. All the other sections are written to prepare for giving the theorem.

----
I updated the score and confidence.


**Limitations And Societal Impact:**

It seems better if this paper gives more background and practical implications to better place this work in the literature.

**Main Review:**

Originality: The task considered here is fresh in LiNGAM and other identifiable causal discovery methods. For nonparametric methods based on conditional independence, there are some works on this consistency issue as on page 2, e.g., Kelly and Mayo-Wilson (2010) and Zhang and Spirtes (2003): Strong Faithfulness and Uniform Consistency in Causal Inference.

Quality: Most of this paper is devoted to proving Theorem 6.1. This would be necessary. Further, this would be better than putting all the proofs in the supplement. On the other hand, when trying to evaluate the value of this paper, I would like to know more background and practical implications.

For example, uniform consistency looks very strong property, and I wonder if how many methods would have uniform consistency. Would well-known statistical methods and machine learning estimation methods, e.g., lasso, have the property? If most of the methods do not have uniform consistency, would it be meaningful in practice to require uniform consistency? Perhaps, Genin and Mayo-Wilson proposed statistical decidability because they thought uniform consistency is a too strong requirement, but point-wise consistency is a too weak requirement.

Kelly and Mayo-Wilson said that the PC algorithm does not have the property in the unconfounded linear Gaussian setting. But, Zhang and Spirtes (2003) seem that they added strong faithfulness to prove the uniform consistency. Also, as in footnote 2, Buhlmann et al. (2014) considered additional assumptions to achieve uniformly consistent procedures.

My question is which consistency concept is good enough when users need to judge which estimations methods are reliable enough. I didn’t fully understand what users can compute and cannot compute for each of the consistency concepts.

In Section 6, they say that it is not possible to construct confidence intervals for causal effects enjoining finite-sample coverage. Does this mean that it is not meaningful to compute confidence intervals for causal effects for other methods than uniformly consistent estimation methods?

In the literature of causal discovery, users often compute bootstrap probabilities of causal orientations, e.g., Komatsu et al. (ICANN2010) and Moneta et al. (2013, Oxford Bulletin of Economics and Statistics). Is this a right procedure for unconfounded LiNGAM but not for confounded LiNGAM?

Other questions are:
1.	whether this consistency property depends on the choice of estimation algorithms, e.g., ICA-based LiNGAM or some other estimation methods for LiNGAM.

2.	Do researchers need to find and add more assumptions to achieve uniform consistency or statistical decidability? Or do researchers need to find some better consistency criterion so that most of the methods satisfy the criterion and the criterion is enough practically useful from some practical viewpoint?

3.	Table 1 summarized the differences between different LiNGAM methods. In this paper, they prove the difference. I’m interested in why and/or how this difference arises.

4. I'm not very familiar with this type of mathematical considerations. For example, In the proof of Lemma 5.2, they consider sequences $X_{1,n}$ and $X_{2,n}$. I didn't follow why considering these sequences is going to prove the lemma. n is the sample size.

Clarity: I felt that the authors did their best to compactly have the relevant theorems and proofs in these eight pages to finally prove the main result, Theorem 6.1. Nevertheless, giving some examples would have helped a lot to better and more quickly understand the content, I think.

Significance: I agree with them that this is the first step in understanding how difficult a problem is, as they stated at the bottom of Section 6. Though most works on causal discovery focus on showing the identifiability conditions and developing efficient algorithms, this type of works would be necessary for users to use causal discovery methods properly.

**Time Spent Reviewing:**

25

---

> ### Author Response · Authors · 2021-08-09
> **Response to Reviewer YVg9**
>
> I am grateful to the reviewer for the careful and generous reading and the helpful and incisive comments. I respond to the comments in the following.
>
> * I would like to know more background and practical implications.
>
> Although the paper focuses on demonstrating the negative result, it’s practical upshot is to suggest several adjustments to the standard assumptions that may make it possible to prove stronger positive results.
>
> The immediate solution that suggests itself is to strengthen the faithfulness assumption so that “near” violations of faithfulness are ruled out. This would block the undecidability result. However, in light of hostile reactions to strong (and weak) faithfulness I think it would be good to explore other avenues.
>
> Alternatively, we could block the undecidability result by strengthening the assumption of non-Gaussianity. I can think of two ways of doing this: (1) rule out “near” Gaussianity; or (2) rule out noise terms with normal components, i.e. noise terms that can be written as X + Z, where Z is Gaussian. Kagan et al. (1973, Thm 10.3.7) use this “no normal component” idea to get stronger uniqueness results in their setting. I suspect this can be exploited in causal discovery as well. Moreover, “no normal components” is only slightly less plausible than non-Gaussianity itself.
>
> All these strengthenings would be sufficient to block the proof strategy used for Lemma 5.2. They may all be sufficient to ensure decidability. However, they would be difficult to motivate if the negative result were not in print someplace. And, as you can see, proving the negative result already results in a paper that is flirting with the page limit.
>
> I also want to emphasize that without the intermediate concept of statistical decidability, we would not have a “target” at which these strengthenings could aim. The standard procedure would be to strengthen assumptions until we get uniform convergence, which may make the assumptions perhaps too strong to be plausible. (See, for example, Caroline Uhler et al. (2013) “Geometry of the faithfulness assumption in causal inference.” in the Annals of Statistics.)
>
> In the revision, I would like to emphasize these positive directions that the submission suggests.
>
> * For example, uniform consistency looks very strong property, and I wonder if how many methods would have uniform consistency. Would well-known statistical methods and machine learning estimation methods, e.g., lasso, have the property? If most of the methods do not have uniform consistency, would it be meaningful in practice to require uniform consistency?
>
> Yes, uniform consistency is provably impossible under standard assumptions (see the example in section 3, Genin and Mayo-Wilson (2020)). That means impossible for *any* method, including lasso and others. Since uniform consistency is impossible, it is indeed not meaningful to insist on it.
>
> * Perhaps, Genin and Mayo-Wilson proposed statistical decidability because they thought uniform consistency is a too strong requirement, but point-wise consistency is a too weak requirement.
>
> Yes, that is exactly how I understand their idea.
>
> * My question is which consistency concept is good enough when users need to judge which estimations methods are reliable enough. I didn’t fully understand what users can compute and cannot compute for each of the consistency concepts.
>
> Of course, uniform consistency is preferable whenever you can get it. If it were possible, a user could compute *a priori* the sample size necessary so that the orientation conclusions of the algorithm would be correct with high probability. However, this is not possible under standard assumptions and assumptions sufficient to ensure uniform consistency are implausibly strong.
>
> When uniform consistency is impossible, one would at least like to have decidability. Although the user could not compute a priori the sample size necessary for a probably correct orientation, the user would be able to recognize from the data when the sample is large enough to make a probably correct orientation. Indeed, as soon as a  decision procedure makes an orientation decision, the sample size must be large enough s.t. the conclusion is probably correct. However, decidability is not possible in the confounded LiNGAM setting.
>
> Lacking uniform consistency and decidability, you would at least like to have decidability in the limit (pointwise consistency). That ensures that *some* sample size is large enough for a probably correct decision, although you would never be able to tell whether your sample is big enough.
>
> * In Section 6, they say that it is not possible to construct confidence intervals for causal effects enjoining finite-sample coverage. Does this mean that it is not meaningful to compute confidence intervals for causal effects for other methods than uniformly consistent estimation methods?
>
> No, confidence intervals are more like the intermediate “decidable” case than the very strong uniform consistency case. If you wanted an a priori sample size by which your confidence interval would be “tight enough”, that would be more like the uniform case.
>
> * In the literature of causal discovery, users often compute bootstrap probabilities of causal orientations, e.g., Komatsu et al. (ICANN2010) and Moneta et al. (2013, Oxford Bulletin of Economics and Statistics). Is this a right procedure for unconfounded LiNGAM but not for confounded LiNGAM?
>
> Yes, under the standard confounded LiNGAM assumptions, these probabilities could not be meaningful.
>
> * whether this consistency property depends on the choice of estimation algorithms, e.g., ICA-based LiNGAM or some other estimation methods for LiNGAM.
>
> No, this is one of the strengths of the submission! It applies to any discovery algorithm.
>
> * Do researchers need to find and add more assumptions to achieve uniform consistency or statistical decidability? Or do researchers need to find some better consistency criterion so that most of the methods satisfy the criterion and the criterion is enough practically useful from some practical viewpoint?
>
> Personally, I think we should try to find nearby assumptions that guarantee decidability but not uniform consistency. The assumptions guaranteeing uniform consistency would be too strong, but I think those guaranteeing decidability would be “just right”. (See also my response to the first comment).
>
> I also think it is a great idea to identify new, better consistency criteria! I would love to see additional suggestions.
>
> * Table 1 summarized the differences between different LiNGAM methods. In this paper, they prove the difference. I’m interested in why and/or how this difference arises.
>
> In a way, the whole thrust of the paper is proving how the difference arises. The kernel of the idea is the example in Figure 3. I am afraid this was not made conspicuous enough. I discuss improvements in this area in my answer to the next comment.
>
> * I'm not very familiar with this type of mathematical considerations. For example, In the proof of Lemma 5.2, they consider sequences $(X_{1,n}, X_{2,n}). $  I didn't follow why considering these sequences is going to prove the lemma. $n$ is the sample size.
>
> The confusion may arise from an overloading of $n$. In this case, $n$ is not sample size but merely the index of the sequence. I will fix this in the revision. I will try to gloss the proof idea more clearly in the revision. What do you think of the following:
>
> We want to prove that even under the assumption of faithfulness, ancestry relationships between observed variables are decidable in the limit, but not decidable. We prove this by showing that you can approximate to an arbitrary degree of precision the distribution over $(X,Y)$ generated by a faithful LiNGAM in which $X\rightarrow Y$ by a series of distributions $(X_m,Y_m)$ generated by faithful but confounded LiNGAMs in which $X_m\leftarrow Y_m$. Since the distributions over the observables can be made arbitrarily similar, there can be no sample size $N$ at which an algorithm could safely output an orientation. Roughly: for every sample size $N$ at which the method orients correctly in $(X,Y),$ there is an $M$ such that the method would orient incorrectly in a nearby distribution $(X_M, Y_M).$
>
> Does that help?
>
> *Nevertheless, giving some examples would have helped a lot to better and more quickly understand the content, I think.
>
> The example in Figure 3 is crucial for understanding the result. If it were better explained, would it be sufficient for understanding the content?
>
> * Significance: I agree with them that this is the first step in understanding how difficult a problem is, as they stated at the bottom of Section 6. Though most works on causal discovery focus on showing the identifiability conditions and developing efficient algorithms, this type of works would be necessary for users to use causal discovery methods properly.
>
> Let me reiterate that I very much appreciate the reviewer’s careful reading and generous comments. I have tried to articulate more clearly the practical upshot of the submission in my response to the first comment. Although it was not intended, the first draft was overly negative. I would like to incorporate this more positive perspective in the revision.

---

> > ### Comment · Reviewer_YVg9 · 2021-08-20
> > **It helped a lot to understand the proof of Lemma 5.2.**
> >
> > Thanks for your feedback! I want to ask some more questions to understand the submission better.
> >
> > 1. We cannot know which property holds from DATA, uniform consistency, pointwise consistency, or statistical decidability. Right? We need to investigate the model we assume to know which of the properties hold. Then, we know what we are allowed to do, e.g., we may compute confidence intervals for models with statistical decidability and those with uniform consistency. Right? What are other things allowed or not allowed to do for models with statistical decidability?
> >
> > 2. To enjoy uniform consistency, we need to know how large sample sizes are enough a priori. To enjoy pointwise consistency (decidability in the limit), we need to have enough large sample sizes, but we don't know how large is large enough. To enjoy statistical decidability,  we also need to have enough large sample sizes, but we have some method that achieves the probability of estimating a wrong model less than alpha for all the sample sizes n? The latter n also needs to be large enough?
> >
> > 3. In the proof of Lemma 5.2, why Z1 and Z2 are taken to be Gaussian? I see that if they were non-Gaussian, then V1 and V2 are not independent, and the lhs model of Fig.3 is not unconfounded LiNGAM anymore. Is this related to your conjecture of ruling out normal components to achieve statistical decidability?

---

> > > ### Author Response · Authors · 2021-08-21
> > > **Second Response to Reviewer YVg9**
> > >
> > > Thank you for your continued thoughtful engagement with the paper. I answer questions in-line.
> > >
> > > * 1. We cannot know which property holds from DATA, uniform consistency, pointwise consistency, or statistical decidability. Right? We need to investigate the model we assume to know which of the properties hold.
> > >
> > > Exactly.
> > >
> > > * Then, we know what we are allowed to do, e.g., we may compute confidence intervals for models with statistical decidability and those with uniform consistency. Right?
> > >
> > > Yes, although the question is a little tricky: we build confidence intervals when we want to estimate real-valued parameters. In this case we are not exactly estimating a real valued parameter but a more qualitative feature (causal orientation) of the structure generating the data. However, a statistical decision procedure is supposed to be an analogue of a procedure for creating confidence intervals. Here is how I understand the analogy:
> > >
> > > If we have a valid and consistent procedure for generating confidence intervals then, although we cannot necessarily say *a priori*  how much data we need to construct an “informative” interval (an interval tight enough to be interesting), we can (1) at every sample size and with high probability construct some interval that contains the true parameter (although it may be true only because it is very wide) and (2) construct tighter and tighter intervals as sample sizes grow.
> > >
> > > If we have a statistical decision procedure then, although we cannot necessarily say *a priori* how much data we need to output an “informative” orientation judgement, we can (1) at every sample size and with high probability output a true orientation judgement (although it may be true only because it is saying something uninformative: “it might be any orientation”) and (2) as sample sizes grow, eventually output an informative judgement.
> > >
> > > * What are other things allowed or not allowed to do for models with statistical decidability?
> > >
> > >
> > > I’m not sure how to answer this, has the preceding helped?
> > >
> > >
> > > * 2. To enjoy uniform consistency, we need to know how large sample sizes are enough a priori.
> > >
> > > Yes. If our problem is solvable with a uniformly consistent method, then we know a priori how large a sample for the method to output a true answer with high probability.
> > >
> > > * To enjoy pointwise consistency (decidability in the limit), we need to have enough large sample sizes, but we don't know how large is large enough.
> > >
> > > Yes. If our problem is solvable only with a pointwise consistent method, then we know that there is some sample size s.t. the outputs of the method are true with high probability, but we do not know how large that sample size is. For any sample size, the output of the method may also be false with high probability. (In other words: how large a sample you need depends on which model is generating the data, which is exactly what you are trying to learn from the data.)
> > >
> > > * To enjoy statistical decidability,  we also need to have enough large sample sizes, but we have some method that achieves the probability of estimating a wrong model less than alpha for all the sample sizes n? The latter n also needs to be large enough?
> > >
> > > Not quite.  Statistical decidability says that *for all sample sizes* the output of the method will be *true* but it will only be informative for *large enough* samples. Obviously, once the sample is large enough, the method will give you a sign by outputting an informative conclusion. At that point you know that you can probably rely on the conclusion, since the method always outputs something true with high probability.
> > >
> > > To sum up:
> > >
> > > If we are in a uniform situation, we know that the output of the method is probably correct and we know a priori how much data we need for it to be informative.
> > >
> > >
> > > If we are in a decidable situation, we know that the output of the method is probably correct, but we don't know a priori how much data we need for it to be informative.
> > >
> > > If we are in an undecidable but pointwise situation, we can never be sure if the output is correct, although we know it will be correct eventually.
> > >
> > > If we don't even have pointwise consistency, we can never be sure if the output is correct, and we do not even know it will be correct eventually.
> > >
> > > * 3. In the proof of Lemma 5.2, why Z1 and Z2 are taken to be Gaussian? I see that if they were non-Gaussian, then V1 and V2 are not independent, and the lhs model of Fig.3 is not unconfounded LiNGAM anymore. Is this related to your conjecture of ruling out normal components to achieve statistical decidability?
> > >
> > > Exactly! I think this trick is relying on an essential property of the Gaussian (see the Lucaks-King or Darmois-Skitovich theorem). I am betting if you didn't have Gaussians to play with you could not pull this off.

---

> > > > ### Comment · Reviewer_YVg9 · 2021-08-22
> > > > **One more question**
> > > >
> > > > Thanks a lot for your reply. Many things become much clear to me!
> > > >
> > > > I have one more question. Would ruling out normal components be something like you assume errors are linear combinations of independent variables and there is no normal variable in the independent variables. I guessed that they need to be independent or of some similar property to exclude non-meaningful cases like e = non-Gaussian + z1 - z1, where z1 is Gaussian. Well, do we need some assumption like the error cannot be linearly decomposed further??

---

> > > > > ### Author Response · Authors · 2021-08-22
> > > > > **normal components**
> > > > >
> > > > > Good question.  Say that an error term E has no normal component if it cannot be expressed as the sum X + Z where X,Z are independent and Z is Gaussian. That rules out the existence of any "longer" linear decomposition (by re-associating the terms in the summation). This requirement is similar, but weaker, than the notion of Indecomposability (see the wiki page on Indecomposable distribution).
> > > > >
> > > > > I was relying on a definition given by Kagan et al. in *Characterization Problems in Mathematical Statistics* (1973, p. 312), but I think they are not careful to state the requirement of independence. The right definition is given by Linnik and Ostrovsky *Decomposition of Random Variables and Vectors* (1977). In the Introduction (p. 1), they point out, as you do, that the notion would be meaningless unless we enforced independence.

---

> > > > > > ### Comment · Reviewer_YVg9 · 2021-08-23
> > > > > > **Thanks a lot! I see.**
> > > > > >
> > > > > > Thanks a lot! I see.

---

### Official Review · Reviewer_xWiG · 2021-07-15

**Rating:** 7
**Confidence:** 3

**Summary:**

This paper focuses on the problem of causal discovery in additive linear models with non-Gaussian errors (LINGAM) and latent variables. It asks: are the underlying orientations statistically decidable (controlled error for every n) for any algorithm? Unlike previous work that answered in the affirmative when all variables are observed, the authors demonstrate that the answer is negative when there is latent confounding. While identifiability guarantees exist in this setting (i.e. causal orientations or equivalently partial orderings are identifiable from data), decidability does not hold, developing further insights about the performance of LINGAM.

**Limitations And Societal Impact:**

None that I can foresee.

**Main Review:**

Originality: to the best of my knowledge, the authors ask and answer an original question about statistical decidability of LINGAM with latent variables. While the proof techniques are quite similar to "Statistical Decidability in Linear, Non-Gaussian Model", there is valuable theoretical understanding that this paper offers.

Quality: After going through some of the main proofs, the technical results appear solid and use neat arguments based on concepts from topology. While I did not understand some of the technical details (see the next paragraph for more discussion on that), I had some potentially naive technical questions for the authors:

-- For uniform consistency, the edge strength should typically be bounded away from zero. Under the same type of assumption, can we obtain statistical decidability in this setting with the latent variables? More generally, would imposing assumptions on the strength of the edges/ number of the latent variables and their effects/DAG structure among observed variables aid with statistical decidability?

--- Lemma 5.2: the argument here is that there is another model with p+2 variables that is also compatible with a LINGAM model. Is the number 2 somehow fundamental? suppose the number of latent variables was only 1. Could a different argument be made to show a similar result? I am sure I am missing some technical understanding here.

-- I recognize that statistical decidability is a desirable attribute and ensures that the "flopping" behavior in causal orientation does not occur. More practically, I wonder how significant this is when $n$ is large where asymptotic consistency kicks in.  In particular, the authors of the paper "Learning Linear Non-Gaussian Causal Models in the Presence of Latent Variables" demonstrate good experimental results of their procedure for identifying causal ordering (While statistical decidability is not satisfied). I just simply wonder, concretely, how relevant this statistical decidability is in practice?


Clarity: my biggest concern with this paper is the amount of notation that the reader must sift through to understand definitions and high-level proof strategy. As somebody who is not an outsider to causal inference with latent variables, I had a very hard time going through this paper, and I imagine most of the readers would have similar difficulties.  Unfortunately, such impenetrability really hurts the paper in my opinion. As such, I think the authors need to make very substantial changes to the writing to improve clarity and readability. Below I outline some suggestions:

--- "prove that if, in addition to
the usual LiNGAM assumptions, we assume causal faithfulness, then causal orientation between
observed variables is identified even in the presence of unobserved confounders." -> perhaps to ensure that this is not misleading for readers, best to say that they prove that the causal ordering is identifiable, even if the causal structure is not. This to me is not the same as saying causal orientation is identified cause one still doesn't know whether there is an edge between two observed variables

--- there is label "34th Causal Discovery & Causality-Inspired Machine Learning Workshop at Neural Information Processing
Systems, 2020" at the bottom of page 1. Please remove.

--- "Causal orientation is statistically decidable,
if for any \alpha> 0; there is a consistent procedure that, at every sample size, hypothesizes a false
orientation with chance less than" -> it's good to define what orientation means right in the introduction. I imagine here a false orientation could also be deciding an orientation when no partial order exists between the variables.

--- Table 1: Causal Orientation in LiNGAM models: it would be good to highlight here what the authors contribution is clearly for quicker readability. Further Table 1 is not referenced in the main text.

--- Since the contribution of this paper is technical, it would be good to state an informal theorem in the introduction and provide a few sentences about the high-level proof technique.

---- "Let Me a set of statistical models. We assume there is a function ....." -> it seems very abrupt to jump into heavy notation without some easing in or at least having a different subsection in the intro.  While i appreciate the precision and careful mathematical definitions, looking at the introduction, it reads as more a mathematics paper than a neurips paper.

---- Statement of Theorem 1.2: a cleaner way to write this it may be:  "for every A,B, P(A) intersect with cl(P(B)) is empty"?

--- Theorem 1.1 and 1.2 appear in the introduction and are not the main result. This seems a bit off. To me, these two theorem statements somehow guide the high-level proof technique. This really does not come through from the writing but is clear later after one reads further in the paper.

---- "A linear causal model in d variables M is a triple": the authors use $d$ here and later $p$. At first, I thought $d$ is the number of observed and latent variables and $p$ is the number of observed variables. This however feels potentially wrong since the random variables are named X. It would be good if the authors describe what is observed and latent as soon as they describe the SCM.

----- The description leading up to the faithful description seems overly complex and unnecessary. I think it would be better to state it that  the model M is faithful if ) The total causal effect from variable X_i
to X_i is nonzero if there is a causal path from X_i to X_j. This translates algebraically to B_ij(M) \neq 0 whenever  j-> i.

----  The paragraph before section 3 introduces more notations. It's best if to condense these and include in the intro. I think the authors should have a notation section and a section with definitions (e.g. faithful, statistical decidability, question, answers,...)

---- Theorem 4.1: how is this different than standard LINGAM result?

---- "That situation is not too worrisome, since the effect of an ideal
intervention on X1 would be the same in both circumstances. What is more worrisome is that
the vector of observed variables O may admit two LiNGAM models that differ on the effects of
interventions." -> what is the definition of intervention here? seems strange to see intervention all of a sudden. Perhaps authors could rephrase.

--- Perhaps some diagrams would help clarify: for example, the authors could show the proof strategy at the beginning of the paper with diagrams indicating the space of questions/answers and the topological arguments that will be made for their proof.

the same of questions and displaying what is needed for statistical decidability.

Significance: To me, while it is nice to have this negative result, I am not sure about the significance of statistical decidability in practice. Perhaps for low sample settings, this is an important question but is likely insignificant in large sample settings. It would help the paper if some empirical demonstrations highlight the relevance. Further, given the large amount of notations in this paper and the current writing structure, I am not sure if this will have lasting impact unless major revisions are conducted.

**Time Spent Reviewing:**

9

---

> ### Author Response · Authors · 2021-08-09
> **Response to Reviewer xWiG**
>
> I am grateful for the reviewer’s careful reading and their thoughtful and incisive suggestions. I abbreviate original comments to meet the character limit.
>
> * For uniform consistency, the edge strength should ... be bounded away from zero. Under the same type of assumption, can we obtain statistical decidability … ? More generally ...
>
> Good question. Bounding edge strengths away from 0 would not help, as the example in Fig. 3 does not rely on making edge strengths arbitrarily small. If you allow for at least 2 latents, I can still run the argument.
>
> What would help is strengthening faithfulness so that “near” violations of faithfulness are ruled out. That would block my proof strategy. But in view of hostile receptions of faithfulness, I suspect many people would find strengthenings of faithfulness implausible.
>
> Another way to block the proof would be to strengthen the assumption of non-Gaussianity. I can think of 2 ways of doing this: (1) rule out “near” Gaussianity; or (2) rule out noise terms with normal components, i.e. noise terms that can be written as X + Z, where Z is Gaussian. Kagan et al. (1973, Thm 10.3.7) use this “no normal component” idea to get stronger uniqueness results in their setting. I suspect this can be exploited in causal discovery as well. Moreover, “no normal components” is only slightly less plausible than non-Gaussianity itself. However, it would be difficult to motivate this condition without the negative result in print someplace.
>
> * Lemma 5.2: the argument here is that there is another model with p+2 variables that is also compatible with a LINGAM model. Is the number 2 ... fundamental?
>
> I wondered this too. So far I have not come up with a way to run the argument with only p+1, but I cannot rule out that it is possible.
>
> * I recognize that statistical decidability is a desirable attribute and ensures that the "flopping" behavior in causal orientation does not occur … I wonder how significant this is when $n$ is large where asymptotic consistency kicks in. … how relevant is this statistical decidability in practice?
>
> This question brings us quickly to the heart of the matter. If we knew *a priori* how large $n$ had to be, we would have uniform convergence. But uniform convergence is not possible under the usual assumptions.
>
> On the other hand, if we could tell from the data when $n$ was large enough, then the question would be statistically decidable: as soon as $n$ is large enough, safely infer an ancestry relationship. But the submission shows that under the standard assumptions, the ancestor relationship is not statistically decidable.
>
> Salehkaleybar et al. show good performance on synthetic data from a chosen DAG. Using the techniques developed in this paper I believe could demonstrate that for any sample size $n$ there is a DAG such that their algorithm is no better than random guessing. As for their results on real data, it is comforting that the results of the algorithm are compatible with common beliefs in economics -- but what if these are wrong? And does it work on other real data?
>
> However, the takeaway of this paper is not that we should give up. I think there are some plausible nearby assumptions that would give us statistical decidability. (See my response to the first comment.) I also want to emphasize that without the intermediate concept of statistical decidability, we would have one less target at which these stronger assumptions could aim. The standard procedure would be to strengthen assumptions until we get uniform convergence. But this makes the assumptions too strong to be plausible. *Aiming for decidability may be the Goldilocks solution.* However, it would be hard to motivate these strengthenings unless the negative result were in print someplace.
>
> * Clarity: my biggest concern with this paper is the amount of notation that the reader must sift through …
>
> I very much regret the notation involved. I struggled to introduce as little as I could, while stating the theorems accurately. I am sure this could have been done better and appreciate suggestions for improvement.
>
> * perhaps to ensure that this is not misleading for readers, best to say that they prove that the causal ordering is identifiable, even if the causal structure is not.
>
> This is a good criticism, but I’m not sure about the proposed solution. To me causal order is ambiguous because it is indifferent to the order between variables that have no relationship either way. Is it better to say that the “ancestry” relationship between observed variables is identified?
>
> *  there is label "34th Causal ..."
>
> Thank you.
>
> * it's good to define what orientation means right in the introduction.
>
> You’re right, I will clarify this in the revision.
>
> * Table 1: Causal Orientation in LiNGAM models: it would be good to highlight here what the authors contribution is clearly for quicker readability …
>
> Good point. I should say that in this paper, what is proven is the top-right corner: although the ancestry relation is decidable in the limit, it is not decidable.
>
> * Since the contribution of this paper is technical, it would be good to state an informal theorem in the introduction and provide a few sentences about the high-level proof technique.
>
> Agreed. How about the following: Even under the assumption of faithfulness, ancestry relationships between observed variables are decidable in the limit, but not decidable. We prove this by showing that you can approximate to an arbitrary degree of precision the distribution over $(X,Y)$ generated by a faithful LiNGAM in which $X\rightarrow Y$ by a series of distributions $(X_n,Y_n)$ generated by faithful but confounded LiNGAMs in which $X_n\leftarrow Y_n$. (In the example in Figure 3, it is a matter of causal orientation and not just ancestry relations.)
>
> * it seems very abrupt to jump into heavy notation without some easing in or at least having a different subsection in the intro.
>
> I will put this in a separate section “Notation”.
>
> *Statement of Theorem 1.2: a cleaner way to write this …
>
> Agreed.
>
> * Theorem 1.1 and 1.2 appear in the introduction and are not the main result. This seems a bit off.
>
> Agreed. I think I should end the intro section with an informal statement of the result and proof idea and then put Theorem 1.1-2 into a subsequent section “Definitions and Prerequisites”.
>
> * "A linear causal model in d variables M is a triple": the authors use $d$  here and later
> $p$. At first, I thought $d$ is the number of observed and latent variables and  $p$ is the number of observed variables. This however feels potentially wrong since the random variables are named X. It would be good if the authors describe what is observed and latent as soon as they describe the SCM.
>
> Yes, on the convention I am using, $d$ is equal to the number of variables (observed or latent) in the SCM generating the data. The variables are called $X_i$. On the other hand, $p$ is the length of the vector of observed variables, each called $O_i$, and identical to some $X_j$. I think it is strange to build in what is observed and what is latent into the SCM. After all, 2 different researchers could measure different variables of the same SCM. I think it is more natural to say they are studying the same SCM, but measuring different variables, than to say they are studying different SCMs. I will be more explicit about this convention in the revision.
>
> * The description leading up to the faithful description seems overly complex and unnecessary.
>
> Agreed, thank you for the simplification.
>
> * The paragraph before section 3 introduces more notations. It's best to condense these and include in the intro.
>
> I agree.
>
> * Theorem 4.1: how is this different than standard LINGAM result?
>
>
> True, it is substantially the same. I found it hard to extract a proof of this from the original 2006 LiNGAM paper, so I prefer the Genin and Mayo-Wilson statement, but I should really say that this is Shimizu et al’s result.
>
> * What is the definition of intervention here? seems strange to see intervention all of a sudden. Perhaps authors could rephrase.
>
> True, I should say: “since the total causal effect of X_1 on X_2 is the same in both circumstances.”
>
> * Perhaps some diagrams would help clarify ...
>
> Good point. I have some ideas for some stylized graphics illustrating the idea. This would be a good use of the additional page of space available if the submission were accepted.
>
> * Significance: To me, while it is nice to have this negative result, I am not sure about the significance of statistical decidability in practice. Perhaps for low sample settings, this is an important question but is likely insignificant in large sample settings.
>
> The paper is written too negatively. Although we don’t have the theoretical resources to say how large a sample is large enough, the emphasis should be on how it shows that some adjustments are necessary to the standard assumptions. Without statistical decidability as a target, it would be hard to come up with plausible adjustments of standard assumptions. In the revision I would like to mention the possible avenues of escape I outlined in my response to the first and third comments.
>
> * It would help the paper if some empirical demonstrations highlight the relevance.
>
>  I think it would be fairly straightforward to show via experiment that standard algorithms would give misleading answers (even at very large samples) in the scenario outlined in Figure 3. I think this would also be a good use of the additional space available to accepted submissions.
>
> *  Further, given the large amount of notations in this paper and the current writing structure, I am not sure if this will have lasting impact unless major revisions are conducted.
>
>  I want to emphasize that I am very grateful for the reviewers effort in reading and critiquing the submission. As a result, the revision will surely be a significant improvement.

---

> > ### Comment · Reviewer_xWiG · 2021-08-22
> > **Response to the author**
> >
> > Thank you for your careful and sincere response to me and the other reviewers, I really appreciate it! I believe your revision will address many of the concerns about the presentation (regarding technicality and style to make it more readable for the general audience). I also believe that your revision will change the tone of the paper to highlight the relevance and potential avenues for future direction, namely standard assumptions (e.g. faithfulness) are not strong enough for decidability and finding stronger assumptions to ensure decidability would be of interest.
> >
> > I do have some remarks/questions and would appreciate your feedback for better understanding:
> >
> > - You hint in the response that their proof for lack of decidability is no longer valid if the error terms in the LINGAM model are sufficiently far from a Gaussian distribution. This of course makes sense to me as in the linear Gaussian case that decidability fails. What is surprising though is that the degree of closeness to Gaussianity is somehow not a problem in LINGAM without hidden confounders. I would have thought that issue would also arise there? Is the issue somehow that the closeness to non-Gaussianity is particularly a problem for the unobserved variables? For example, if the assumption was that the hidden confounders are far from Gaussian, then, the noise terms for the observed variables can be arbitrarily close to a Gaussian without sacrificing decidability?
> >
> > - To clarify my understanding, suppose we had access to a statistically decidable procedure. Then, for every $n$, we would get a reliable "answer". For $n$ not large enough, this reliable answer may say that there is not enough evidence to decide on one direction over another. For large enough $n$, the reliable answer would say that a certain direction can be trusted with high probability. Is my understanding correct? If so, and assuming a procedure is decidable, could one in practice do some sort of subsampling to figure out the informativeness of the output? If a certain edge is selected across many subsamples, then one could trust this as a discovery. Otherwise, the output could be that there is not enough samples to reliably make an informative statement. If only point wise convergence held, one is not able to make any such statements, correct?
> >
> > - Looking at the future work, you point out that stronger assumptions may yield decidability. Of course, these assumptions should be weak enough to not already guarantee uniform convergence. While the paper of Genin and Mayo-Wilson (LINGAM without hidden confounders) propose that the LINGAM model is decidable, to best of my understanding, they cannot show concrete algorithms that attain this decidability. I suspect many of the same challenges would appear with this paper's setting. These are of course all interesting directions for future research but maybe much more needs to be understood for this notion to have practical use? Ultimately,  the goal is to show a particular algorithm attains statistical decidability?

---

> > > ### Author Response · Authors · 2021-08-23
> > > **Response to Reviewer xWiG**
> > >
> > > * Thank you for your careful and sincere response to me and the other reviewers, I really appreciate it! I believe your revision will address many of the concerns about the presentation (regarding technicality and style to make it more readable for the general audience). I also believe that your revision will change the tone of the paper to highlight the relevance and potential avenues for future direction, namely standard assumptions (e.g. faithfulness) are not strong enough for decidability and finding stronger assumptions to ensure decidability would be of interest.
> > >
> > > Thank you for your continued thoughtful engagement with the submission! I respond to questions in-line.
> > >
> > >
> > > * You hint in the response that their proof for lack of decidability is no longer valid if the error terms in the LINGAM model are sufficiently far from a Gaussian distribution. This of course makes sense to me as in the linear Gaussian case that decidability fails. What is surprising though is that the degree of closeness to Gaussianity is somehow not a problem in LINGAM without hidden confounders. I would have thought that issue would also arise there? Is the issue somehow that the closeness to non-Gaussianity is particularly a problem for the unobserved variables? For example, if the assumption was that the hidden confounders are far from Gaussian, then, the noise terms for the observed variables can be arbitrarily close to a Gaussian without sacrificing decidability?
> > >
> > > It is a good question and I do not have a complete answer. Assuming that the error terms are bounded away from Gaussianity certainly blocks the non-decidability proof strategy in the submission, but at the moment I do not know how to prove that there is no other way to do it. The provisional explanation I am giving to myself is that, in the unconfounded case, near-Gaussianity blocks uniform convergence but, in the confounded case, it blocks decidability.
> > >
> > >
> > > * To clarify my understanding, suppose we had access to a statistically decidable procedure. Then, for every $n$, we would get a reliable "answer". For $n$ not large enough, this reliable answer may say that there is not enough evidence to decide on one direction over another. For large enough $n$, the reliable answer would say that a certain direction can be trusted with high probability. Is my understanding correct?
> > >
> > > Yes. To sum up decidability and how it relates to the other notions:
> > >
> > > If we are in a uniform situation, we know that the output of a good method is probably correct and we know a priori how much data we need for it to be informative.
> > >
> > >
> > > If we are in a decidable situation, we know that the output of a good method is probably correct, but we don't know a priori how much data we need for it to be informative.
> > >
> > > If we are in an undecidable but pointwise situation, we have no bound on the probability that (even a good) method is incorrect, although we know it will be correct eventually.
> > >
> > >
> > > * If so, and assuming a procedure is decidable, could one in practice do some sort of subsampling to figure out the informativeness of the output? If a certain edge is selected across many subsamples, then one could trust this as a discovery. Otherwise, the output could be that there is not enough samples to reliably make an informative statement. If only point wise convergence held, one is not able to make any such statements, correct?
> > >
> > > Apologies for insisting on a pedantic distinction here. Statistical problems (or “questions”), rather than procedures, are uniformly decidable/decidable/decidable in the limit. If the problem you are facing is decidable, then there is a decision procedure with the relevant trustworthiness property. If you are in possession of such a decision procedure, there is no reason to do subsampling. On the other hand, if the problem you are dealing with is not decidable, then there is no way of subsampling your way out of this difficulty.
> > >
> > > But I think your suggestion is different: if our problem is decidable, can we “boost” a pointwise consistent method into a decision procedure by subsampling? This is an intriguing suggestion and may work in practice. But since limiting decidability is such a weak notion, there is no way it could work in general. For example, take any limiting decision procedure and replace it with the same procedure except that for sample sizes under 1000, it always says “X causes Y” irrespective of the data. Since this procedure cleans up its act after sample size 1000, it is a limiting decision procedure. But for sample sizes <1000, subsampling will always give you the same answer, so it wont turn it into a decision procedure.
> > >
> > > So how do we construct decision procedures? There is a general recipe, but I do not claim that you would want to follow it in every case. The results of Genin and Kelly (2017) imply that if a question {A_1, A_2} is decidable then there is a consistent hypothesis test with bounds on Type I error at every sample size of A_1 against A_2 and vice versa. The following always yields a decision procedure: test A_1 against A_2 and vice versa. If no test, or both tests, reject, then suspend judgement. If only the test of A_1 rejects, conjecture A_2. If only the test of A_2 rejects, conjecture A_1.
> > >
> > > * Looking at the future work, you point out that stronger assumptions may yield decidability. Of course, these assumptions should be weak enough to not already guarantee uniform convergence. While the paper of Genin and Mayo-Wilson (LINGAM without hidden confounders) propose that the LINGAM model is decidable, to best of my understanding, they cannot show concrete algorithms that attain this decidability. I suspect many of the same challenges would appear with this paper's setting. These are of course all interesting directions for future research but maybe much more needs to be understood for this notion to have practical use? Ultimately,  the goal is to show a particular algorithm attains statistical decidability?
> > >
> > > Yes, that is the ultimate goal! This work is like complexity theory in CS: it is meant to tell you how hard a problem is and, therefore, what kind of performance we should expect from a method designed to solve it. In general I think you should always do the best you can, but we don’t know whether a method is doing the best it can until we know how difficult the problem is. This paper is trying to answer that latter question, with a view to developing methods that have the best possible performance. I do not know how difficult it will be to construct methods that exhibit the best performance, but I think if these standards of good performance gain any traction in the causal discovery community the problem can be solved pretty quickly.

---

> > > > ### Comment · Reviewer_xWiG · 2021-08-27
> > > > **this helps, thank you!**
> > > >
> > > > Sorry for my delayed response. Than you for the clarifications! Things are making a lot more sense. I would like to ask one additional clarification with the point you made (sorry I am dragging this on a bit. I am asking this out of curiosity because I find the ideas interesting)
> > > >
> > > > "If the problem you are facing is decidable, then there is a decision procedure with the relevant trustworthiness property. If you are in possession of such a decision procedure, there is no reason to do subsampling"
> > > >
> > > > Suppose we are in a low sample setting and we have a decidable algorithm. Then the algorithm may claim it is not able to differentiate X -> Y and Y -> X. Nonetheless, the truth is going to lie in this output. On the other hand, for a large sample size, the algorithm would claim X-> Y. Then, we would be able to conclude that this is a correct orientation with high probability. Is this correct?

---

> > > > > ### Author Response · Authors · 2021-08-28
> > > > > **re: this helps, thank you!**
> > > > >
> > > > > Thank you for the continued engagement!
> > > > >
> > > > > * Suppose we are in a low sample setting and we have a decidable algorithm. Then the algorithm may claim it is not able to differentiate X -> Y and Y -> X. Nonetheless, the truth is going to lie in this output.
> > > > >
> > > > > Yes, exactly.
> > > > >
> > > > > * On the other hand, for a large sample size, the algorithm would claim X-> Y. Then, we would be able to conclude that this is a correct orientation with high probability. Is this correct?
> > > > >
> > > > > Yes. If I were to nitpick, I would say: if the decision procedure has a greater than $\epsilon$ chance of claiming that X->Y, then in fact X->Y (or some model assumption failed). Whether we can infer from this feature that the post-sample output of the method is correct with high probability depends on your philosophy of probability. A strict frequentist might say we cannot: "the causal facts are not determined by a chance process, so the probability that X->Y is true is either zero or one."
> > > > >
> > > > > PS This is similar to the situation with confidence intervals. If my interval-generating procedure has the confidence property then, with 95% pre-sample probability, it outputs an interval that contains the true parameter. But a strict frequentist would rap my knuckles if I concluded post-sample from a particular confidence interval that the interval contained the true parameter with high probability. I can hear my old stats prof saying "the parameter value is not determined by a chance process, so the probability it is in this *particular* interval is either zero or one."

---

> > > > > > ### Comment · Reviewer_xWiG · 2021-09-09
> > > > > > **Thank you for all the clarifications/help**
> > > > > >
> > > > > > I very much appreciate it! I learned a lot during this process.

---

### Official Review · Reviewer_YEWp · 2021-07-16

**Rating:** 6
**Confidence:** 1

**Summary:**

This paper focuses on the consistency of the causal discovery algorithms especially focusing on the LiNGAM with latent variables. The authors show the direction of any causal edge in a confounded LiNGAM is not statistically decidable.

**Limitations And Societal Impact:**

Yes

**Main Review:**

The paper analyzes the statistical consistency of the causal discovery method.
The paper builds on the decidability of the LiNGAM.  The authors put some effort into relaxing the sufficiency assumption and allowing latent confounders.
The theoretical contributions of this paper are both novel and significant.

However, the submitted manuscript does not use the NeurIPS 2021 LaTeX style file and adds the appendix into the main paper.  I tend to reject this paper.


I might have some misunderstandings about this theme or Formatting rules. So it is very likely that I will change my score if the other reviewers provide some new points.

**Time Spent Reviewing:**

3 hours

---

> ### Author Response · Authors · 2021-08-09
> **Response to Reviewer YEWp**
>
> I am grateful for the reviewers attention to this submission. Needless to say, any formatting and style issues would be resolved in a revision. The mistakes made in meeting the style guidelines were unintentional and were certainly not made in an attempt to defeat guidelines on page limits. I believe the submission could easily be made to conform with the style guidelines.
>
> In case it’s relevant,  I think the reason I missed the guidelines on supplementary material is because I was looking at the "Supplementary material" section in the CFP (https://nips.cc/Conferences/2021/CallForPapers) which doesn't mention that supplementary material has to be in a separate file. I didn't realize there were more instructions about supplementary materials in the page for style files (https://neurips.cc/Conferences/2021/PaperInformation/StyleFiles). I understand that reviewers are under no obligation to look at the appendix, but I thought it would be convenient for the reviewers if everything was in one place. This is my first submission to NeurIPS, so conventions that may be obvious to others are new to me.
>
> The reviewer kindly states that the theoretical contributions of the paper are both novel and significant. I would very much like to know how the reviewer would rate the submission were it not for the stylistic errors. Once again I am grateful for the reviewers time and attention to the submission and I want to reiterate that a revision could easily be made to conform with style guidelines.

---

> > ### Comment · Reviewer_YEWp · 2021-09-05
> > **Feedback after author response**
> >
> > I appreciate the author's clarifications. I will increase my score to a 6 conditional on the authors addressing the concerns noted by all the other reviewers.

---

### Official Review · Reviewer_oGNr · 2021-07-19

**Rating:** 6
**Confidence:** 3

**Summary:**

The paper shows that for linear models with additive non-Gaussian noise components (LiNGAM) in the presence of latent confounders the inference of the direction of causal relations converges in the limit in probability to the correct orientation, but no longer satisfies the criterion for ‘statistical decidability’, this in contrast to unconfounded LiNGAMs. This result could have implications for the interpretation of causal models discovered under the LiNGAM framework.

**Ethical Concerns:**

No ethical concerns

**Limitations And Societal Impact:**

No negative societal impact.

**Main Review:**

High quality paper on a rather technical subject, but relevant to the theoretical foundations of causal inference in LiNGAM models with confounding. The main part of the paper is used to introduce/define various notions of decidability and identifiability, and to discuss existing results for linear additive noise models. The authors then construct a sequence of faithful LiNGAM models with confounding that come arbitrarily close in distribution to unfaithful linear Gaussian models, which we know are not decidable.

The key argument is described in p7, Fig3/Lemma 5.2. This in itself is an interesting example, but I am a bit concerned about the interpretation of the combined ‘convergence in probability’ but at the same time claiming ‘but not quite Gaussian/unfaithful’, even though it obviously gets arbitrarily close to Gaussian/unfaithful. Though perhaps technically correct, this seems more to do with a slight mismatch between the statistical definition of ‘convergence in probability’ and our standard causal concepts of faithfulness and non-Gaussianity, that allows to construct sequences of models that become arbitrarily close to unfaithful / Gaussian (but not quite) whereas they do satisfy the notion of ‘convergence in probability’.

In other words: to me this is not so much a stark warning or novel insight into differing behaviour / properties between LiNGAM models with and without confounders, as perhaps an argument that our intuitive notion of faithfulness needs to be refined/strengthened a bit to avoid this sort of technical discrepancies in the limit.

Apart from that I would suggest the authors specify the current result only applies to acyclic LiNGAMs, as cyclic (non-recursive) models also qualify as LiNGAM, but some of the proof steps specifically rely on lower triangular coefficient matrices (although I am sure the result can be extended to the cyclic case as well).

All in all a good paper, though perhaps a bit too niche & technical. I am not entirely sure this would make it a good fit for the conference: perhaps a more statistically oriented venue would be more appropriate, though I’m sure it will find an audience at NeurIPS too. The main conclusion is true, but the impact will be limited as it is primarily a theoretical issue for a very specific case of causal inference, and I do not consider it a crucial new insight or important caveat on the reliability of statistical causal discovery with latent confounders in general. However it could make a case for the need to refine our notion of faithfulness in the limit.

=> For now: borderline to weak accept, but with medium confidence as I am not an expert in this particular area.

- originality: novel and interesting example that supports the core argument,
- quality: high quality
- clarity: very technical, but well written and still fairly good to follow for a non-insider
- significance: interesting find, but not sure it will have much impact beyond statistical & causal purists

minor comments:
- why no line numbers?
introduction: ‘causal orientation is often not identified’ => this suggestion is wrong. for sparse graphs typically the (vast) majority are identifiable in the equivalence class, even for large graphs with or without confounding
p2,top: ‘uniform consistency’ => here a ref. to ‘Uniform Consistency in Causal Inference’ (Robins,Scheines,Spirtes,Wasserman 2003) would be appropriate
p2,mid: ‘locally closed’ : so an open set is still ‘locally closed’ if it can be contained in a closed set?
p3,start sec.2: change to ‘An *acyclic* linear causal model’ (or ‘recursive’)
idem: would the results in this paper change if M is allowed to be non-recursive?
p3, term (1.): what is the role of \omega in this equation?
p5,Lerm/Thm/Cor 3.1: clear and interesting results, though not a fan of overloaded label numbers :)
p6,Fig3: mention the intrinsic noise term \epsilon_1+3 have switched places (I did not notice this at first)
p7,Fig3/Lemma 5.2: interesting example, but see main review
p8,below Thm 5.1: ‘consequences of Lemma 5.1’ => I assume you mean ‘Theorem 5.1’ right?
p9, mid: I feel the conclusions in this section are a bit too negative in the sense that this should not be a fundamental warning on the statistical reliability of causal inference in the presence of confounding, but more a motivation to strengthen / formalise our intuitive notion of faithfulness

**Time Spent Reviewing:**

3.5

---

> ### Author Response · Authors · 2021-08-09
> **Response to Reviewer oGNr**
>
> I am very grateful to the reviewer for their careful reading and their fair and constructive response to the submission. I very much agree that we ought not to let this result discourage us from causal discovery in the potentially confounded setting. I did not intend this “dismal” interpretation, although some of the prose strikes me now as rather downbeat. This could be easily remedied in the revision. I propose that the submission suggests several adjustments of our assumptions that could lead to more positive results. I also want to emphasize that without the intermediate concept of statistical decidability, we would have one less target at which these adjustments could aim. The standard procedure would be to strengthen assumptions until we get uniform convergence, which may make the assumptions perhaps too strong to be plausible. (See, for example, Caroline Uhler et al. (2013) “Geometry of the faithfulness assumption in causal inference.” in the Annals of Statistics.)
>
> First, as the reviewer suggests, we could strengthen the faithfulness assumption. It is interesting if there is a plausible strengthening of faithfulness that ensures decidability but falls short of ensuring uniform convergence. I think this is a very good direction for future work, but in light of hostile reactions to strong (and weak) faithfulness I think it would also be good to explore other avenues.
>
> Second, we could strengthen the assumption of non-Gaussianity in several ways. One idea is to bound noise terms away from Gaussianity. Another idea is to assume that noise terms have no Gaussian component, i.e. they cannot be written as X + Z, where Z is Gaussian. Kagan et al. (1973) use this “no normal component” idea to get stronger uniqueness results in their setting. I suspect this strategy can be put to use in causal discovery as well. I think it also has the advantage of only being slightly less plausible than the assumption of non-Gaussianity itself.
>
> All these strengthenings would be sufficient to block the proof strategy used for Lemma 5.2. They may all be sufficient to ensure decidability. However, they would be difficult to motivate if the negative result were not in print someplace. And, as you can see, proving the negative result already results in a paper that is flirting with the page limit.
>
> I want to thank the reviewer again for their extremely incisive and helpful response. I respond to individual comments below.
>
>
> Main Review:
> * The authors then construct a sequence of faithful LiNGAM models with confounding that come arbitrarily close in distribution to unfaithful linear Gaussian models, which we know are not decidable.
>
> This is an accurate description, but the argument strategy does not rely on previous knowledge about unidentifiability in the linear Gaussian setting. Rather, it shows that a series of faithful but confounded LiNGAMs where $X_{1,n} \leftarrow X_{2,n}$ generate distributions over the observed variables that converges to the distribution generated by a faithful but confounded LiNGAM where $X_1 \rightarrow X_2.$ In English: although you cannot create a “perfect” illusion (one where the distributions over the observables are literally identical), you can approximate arbitrarily well a faithful confounded LiNGAM with one causal orientation by a series of faithful but confounded LiNGAMs with the opposite orientation.
>
> * In other words: to me this is not so much a stark warning or novel insight into differing behaviour / properties between LiNGAM models with and without confounders, as perhaps an argument that our intuitive notion of faithfulness needs to be refined/strengthened a bit to avoid this sort of technical discrepancies in the limit.
>
> I do not have a single lesson that I want the readers to take away. One good response is to find plausible nearby assumptions that guarantee decidability. However, I think it helps to have decidability as a target! Previously, it would have been natural to look for assumptions that guarantee uniform convergence, which is strictly stronger than decidability. There are reasons to think that such assumptions are implausibly strong. (See Caroline Uhler et al. (2013) “Geometry of the faithfulness assumption in causal inference.”) Perhaps aiming for decidability is the Goldilocks solution. That might ultimately be the lasting contribution of this work (if any) and assuming some strengthened version of faithfulness may accomplish this. (Note however that it might also be sufficient to keep things bounded away from Gaussianity; or to assume that the noise terms have no “Gaussian components” i.e.that the noise term cannot be factored into X + Z, where Z is Gaussian. ) It is an open question for me and I think it would be an interesting direction for future work!
>
> * Apart from that I would suggest the authors specify the current result only applies to acyclic LiNGAMs, as cyclic (non-recursive) models also qualify as LiNGAM, but some of the proof steps specifically rely on lower triangular coefficient matrices (although I am sure the result can be extended to the cyclic case as well).
>
> Thank you, this is a good catch. In the next version I will make sure to track down where acyclicity is actually being used and state the assumption more clearly.
>
> * All in all a good paper, though perhaps a bit too niche & technical. I am not entirely sure this would make it a good fit for the conference: perhaps a more statistically oriented venue would be more appropriate, though I’m sure it will find an audience at NeurIPS too.
>
> Fair point. Much of the work I am responding to (Shimizu et al.’s original 2006 paper; as well as Salehkaleybar et al.’s recent (2020) paper on identifiability in the confounded LiNGAM setting) is published in the JMLR. Is this a more technical venue? I have not applied to NeurIPS before and assumed the two venues were roughly similar.
>
> Minor Comments:
>
> * why no line numbers?
>
> This was not intentional. I will be sure to fix it in the next version.
>
> * introduction: ‘causal orientation is often not identified’ => this suggestion is wrong. for sparse graphs typically the (vast) majority are identifiable in the equivalence class, even for large graphs with or without confounding
>
> Thank you. Is there a standard reference for this?
>
> * p2,top: ‘uniform consistency’ => here a ref. to ‘Uniform Consistency in Causal Inference’ (Robins,Scheines,Spirtes,Wasserman 2003) would be appropriate .
>
> Good catch. I usually cite this paper and will include it in the next version.
>
> * p2,mid: ‘locally closed’ : so an open set is still ‘locally closed’ if it can be contained in a closed set?
>
> A locally closed set is any set that can be written as an intersection of an open and a closed set. Since the set of all distributions is both open and closed, every open and closed set can be written this way (as the intersection of itself with the universal set). So the locally closed sets include all the open and closed sets. I should add a sentence clarifying that the locally closed sets include all the open and closed sets but there are “properly” locally closed sets that are neither open nor closed.
>
> * p3,start sec.2: change to ‘An acyclic linear causal model’ (or ‘recursive’) idem: would the results in this paper change if M is allowed to be non-recursive?
>
> I will update to include acyclicity. I am grateful to the reviewer for suggesting this generalization -- I too suspect that the results can be generalized to the cyclic case. However, making sure this works would require close attention to the technical details and I would not be sufficiently confident about the results to update the paper for the next round.
>
> * p3, term (1.): what is the role of \omega in this equation?
>
> It can be omitted. It is emphasizing that all the random variables are functions from the same probability space. That was helpful for me in thinking about the LiNGAM setup, but is probably just confusing for the reader.
>
> * p5,Lerm/Thm/Cor 3.1: clear and interesting results, though not a fan of overloaded label numbers :)
>
> Thank you. Should the Lemmas/Corollaries be promoted to Theorems? Or should all Lemmas, Theorems and Corollaries use the same counter?
>
> * p6,Fig3: mention the intrinsic noise term \epsilon_1+3 have switched places (I did not notice this at first)
>
> A helpful suggestion --- thank you.
>
> * p7,Fig3/Lemma 5.2: interesting example, but see main review.
>
> OK, see response to main review.
>
> * p8,below Thm 5.1: ‘consequences of Lemma 5.1’ => I assume you mean ‘Theorem 5.1’ right?
>
> Yes. Thank you!
>
>  * p9, mid: I feel the conclusions in this section are a bit too negative in the sense that this should not be a fundamental warning on the statistical reliability of causal inference in the presence of confounding, but more a motivation to strengthen / formalise our intuitive notion of faithfulness
>
> This is really how I meant to conclude, but on second reading I agree it sounds more negative than intended. I think the paper suggests several plausible strengthenings. (1) is to strengthen faithfulness. I think this route is fairly well-explored and --- in view of objections to strong (and weak) faithfulness -- would in my opinion be good to avoid if possible. Another idea is to strengthen the assumption of non-Gaussianity either by (2) assuming that noise terms are in some sense bounded away from Gaussianity or (3) by assuming that noise terms have no Gaussian components, i.e. they cannot be written as X + Z, where Z is Gaussian. I think the latter idea is the most promising, because there exist theorems in Kagan et al. (1973) suggesting that you would get stronger results this way (see their Theorem 10.3.7). I think if non-Gaussianity is a plausible assumption then so is the assumption of no Gaussian components. This would be the route I would like to go, but it would be hard to motivate without having a negative result in print someplace. As you can see, the paper is already flirting with the page limit.

---

### Decision · Program_Chairs · 2021-09-27

**Decision:**

Accept (Poster)

**Comment:**

A nice summary of this paper from one of the reviews:

"The paper shows that for linear models with additive non-Gaussian noise components (LiNGAM) in the presence of latent confounders the inference of the direction of causal relations converges in the limit in probability to the correct orientation, but no longer satisfies the criterion for ‘statistical decidability’, this in contrast to unconfounded LiNGAMs. This result could have implications for the interpretation of causal models discovered under the LiNGAM framework."

While initially the reviewer opinion was mixed, the discussion with the authors led to an emerging consensus that the paper is novel, interesting, and a worthwhile addition to the NeurIPS proceedings.